# Accelerating Exploration with Unlabeled Prior Data

**Qiyang Li**[αγ*]**, Jason Zhang**[α*]**, Dibya Ghosh**[α]**, Amy Zhang**[βγ]**, Sergey Levine**[α]
UC Berkeley[α], UT Austin[β], Meta[γ]
{qcli,jason.z,dibya.ghosh}@berkeley.edu
amy.zhang@austin.utexas.edu, svlevine@eecs.berkeley.edu

## Abstract

Learning to solve tasks from a sparse reward signal is a major challenge for standard reinforcement learning (RL) algorithms. However, in the real world, agents rarely need to solve sparse reward tasks entirely from scratch. More often, we might possess prior experience to draw on that provides considerable guidance about which actions and outcomes are possible in the world, which we can use to explore more effectively for new tasks. In this work, we study how prior data without reward labels may be used to guide and accelerate exploration for an agent solving a new sparse reward task. We propose a simple approach that learns a reward model from online experience, labels the unlabeled prior data with optimistic rewards, and then uses it concurrently alongside the online data for downstream policy and critic optimization. This general formula leads to rapid exploration in several challenging sparse-reward domains where tabula rasa exploration is insufficient, including the AntMaze domain, Adroit hand manipulation domain, and a visual simulated robotic manipulation domain. Our results highlight the ease of incorporating unlabeled prior data into existing online RL algorithms, and the (perhaps surprising) effectiveness of doing so.

## 1 Introduction

Exploration, particularly in sparse reward environments, presents a major challenge for reinforcement learning, and standard exploration methods typically need to seek out all potentially novel states to cover all the places where high rewards may be located. Luckily, in many real-world RL settings, it is straightforward to obtain prior data that can help the agent understand how the world works. For example, if we are trying to find where we left our keys, we would not relearn how to navigate our environment, but rather might revisit locations that we recall from memory. If the data has reward annotations, pretraining with offline RL provides one solution to accelerate online finetuning. However, in many domains of interest, it is more likely for prior datasets to be task-agnostic, and not be labeled with the reward for the new task we hope to solve. Despite not having reward labels, such prior data can be useful for an agent exploring for a new task since it describes the environment dynamics and indicates regions of the state space that may be interesting to explore.

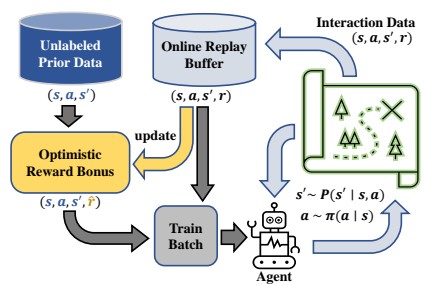

Figure 1: **How to leverage unlabeled prior data for efficient online exploration?** We label the prior data with an optimistic reward estimate and run RL on both the online and offline data. This allows more efficient exploration around the trajectories in the prior data, improving sample efficiency especially for hard exploration tasks.

---

*Equal Contribution

37th Conference on Neural Information Processing Systems (NeurIPS 2023).

How can this unlabeled data be utilized by a deep RL algorithm? Prior data informs the agent about potentially high-reward or high-information states, and behaviors to reach them. Initially, the agent may act to reach these states, leading to directed exploration in a more informative subspace rather than the complete state space. This should be an iterative process, since the agent can use the new online experience to refine what directions from the prior data it chooses to explore in. After the agent begins to receive reward signals, the prior data can help the agent to more reliably reach and exploit the reward signals, steering the exploration towards a more promising region.

Perhaps surprisingly, such informed exploration can be elicited in a very simple way: by labeling the prior data with *optimistic* rewards and adding it to the online experience of an off-policy RL algorithm (Figure 1). Specifically, we choose to impute the missing reward on the unlabeled prior data with an optimistic upper-confidence bound (UCB) of the reward estimates obtained through the agent's collected online experience. Early in training, these optimistic reward labels will be high for states from the prior data, and optimizing the RL objective with this data will elicit policies that attempt to reach regions of state present in the prior data. As states in the prior data are visited, the labeled reward for these states will regress towards their ground-truth values, leading to a policy that either explores other states from the prior data (when true reward is low) or returns to the region more consistently (when high). By training an RL policy with a UCB estimate of the reward, exploration is guided by both task rewards and prior data, focusing only on visiting states from the prior data that may have high reward. In practice, we find that this general recipe works with different off-policy algorithms [Ball et al., 2023, Kostrikov et al., 2021].

Our main contribution is a simple approach to leverage unlabeled prior data to improve online exploration and sample efficiency, particularly for sparse long-horizon tasks. Our empirical evaluations, conducted over domains with different observation modalities (e.g., states and images), such as simulated robot navigation and arm and hand manipulation, show that our simple optimistic reward labeling strategy can utilize the unlabeled prior data effectively, often as well as the prior best approach that has access to the same prior data with *labeled rewards*. In addition, we can leverage offline pre-training of task-agnostic value functions (e.g., ICVF [Ghosh et al., 2023]) from offline data to initialize our optimistic reward model to further boost the online sample efficiency in more complex settings, such as image-based environments. In contrast to other approaches in the same setting, our method is conceptually simpler and easier to be integrated into current off-policy online RL algorithms, and performs better as we will show in our experiments.

## 2    Problem Formulation

We formalize our problem in an infinite-horizon Markov decision process (MDP), defined as $\mathcal{M} = (\mathcal{S}, \mathcal{A}, \boldsymbol{P}, r, \gamma, \rho)$. It consists of a state space $\mathcal{S}$, an action space $\mathcal{A}$, transition dynamics $\boldsymbol{P} : \mathcal{S} \times \mathcal{A} \mapsto \mathcal{P}(S)$, a reward function $r : \mathcal{S} \times \mathcal{A} \mapsto \mathbb{R}$, a discount factor $\gamma \in (0, 1]$, an initial state distribution of interest $\rho$. Algorithms interact in the MDP online to learn policies $\pi(a|s) : \mathcal{S} \mapsto \mathcal{P}(\mathcal{A})$ that maximize the discounted return under the policy $\eta(\pi) = \mathbb{E}_{s_{t+1} \sim \boldsymbol{P}(\cdot|s_t, a_t), a_t \sim \pi(\cdot|s_t), s_0 \sim \rho} \left[ \sum_{t=0}^{\infty} \gamma^t r(s_t, a_t) \right]$. We presume access to a dataset of transitions with *no reward labels*, $\mathcal{D} = \{(s_i, a_i, s_i')\}_i$ that was previously collected from $\mathcal{M}$, for example by a random policy, human demonstrator, or an agent solving a different task in the same environment. The core challenge in this problem setting is to explore in a sparse reward environment where you do not know where the reward is. While structurally similar to offline RL or online RL with prior data, where the agent receives a reward-labelled dataset, the reward-less setting poses a significant exploration challenge not present when reward labels are available. Typically offline RL algorithms must find the best way to use the prior data to *exploit* the reward signal. In our setting, algorithms must use the prior data to improve *exploration* to acquire a reward signal. Using this prior data requires bridging the misalignment between the offline prior dataset (no reward) and the online collected data (contains reward), since standard model-free RL algorithms cannot directly use experiential data that has no reward labels.

## 3    Exploration from Prior Data by Labeling Optimistic Reward (EXPLORE)

In this section, we detail a simple way to use the unlabeled prior data directly in a standard online RL algorithm. Our general approach will be to label the reward-free data with learned optimistic

rewards and include this generated experience into the buffer of an off-policy algorithm. When these rewards are correctly chosen, this will encourage the RL agent to explore along directions that were present in the prior data, focusing on states perceived to have high rewards or not yet seen in the agent's online experience.

---

**Algorithm 1** EXPLORE

---
1: **Input:** prior unlabeled data $\mathcal{D}_{\text{offline}}$
2: Initialize the UCB estimate of the reward function: $\text{UCBR}(s, a)$
3: Online replay buffer $\mathcal{D} \leftarrow \emptyset$
4: Initialize off-policy RL algorithm with a policy $\pi$.
5: **for** each environment step **do**
6:     Collect $(s, a, s', r)$ using the policy $\pi$ and add it to the replay buffer $\mathcal{D}$.
7:     Update the UCB estimate $\text{UCBR}(s, a)$ using the new transition $(s, a, s', r)$
8:     Relabel each transition $(s, a, s')$ in $\mathcal{D}_{\text{offline}}$ with $\hat{r} = \text{UCBR}(s, a)$
9:     Run off-policy RL update on both $\mathcal{D}$ and the relabeled $\mathcal{D}_{\text{offline}}$ to improve the policy $\pi$.
10: **end for**

---

**General algorithm.** Since the prior data contains no reward labels, the agent can only acquire information about the reward function from the online experience it collects. Specifically, at any stage of training, the collected online experience $\mathcal{D}$ informs some posterior or confidence set over the true reward function that it must optimize. To make use of the prior data, we label it with an upper confidence bound (UCB) estimate of the true reward function. Optimizing an RL algorithm with this relabeled prior experience results in a policy that is "optimistic in the face of uncertainty", choosing to guide its exploration towards prior data states either where it knows the reward to be high, and prior data states where it is highly uncertain about the reward. Unlike standard exploration with optimistic rewards for states seen online, the prior data may include states that are further away than those seen online, or may exclude states that are less important. Consequently, the RL agent seeks to visit regions in the prior data that are promising or has not visited before, accelerating online learning.

To explore with prior data, our algorithm maintains an uncertainty model of the reward, to be used for UCB estimates, and an off-policy agent that trains jointly on relabelled prior data and online experience. As new experience is added through online data collection, we use this data to update the agent's uncertainty model over rewards. When training the off-policy agent, we sample data from both the online replay buffer (already reward labeled) and from the prior data (no reward labels). The prior data is labelled with rewards from the UCB estimate of our uncertainty model, and then the online and prior data is jointly trained upon. Training with this data leads to a policy that acts optimistically with respect to the agent's uncertainty about the reward function, which we use to guide exploration and collect new transitions. Algorithm 1 presents a sketch of our method.

Notice that while this strategy is similar to exploration algorithms that use novelty-based reward bonuses [Bellemare et al., 2016, Tang et al., 2017, Pathak et al., 2017, Burda et al., 2018, Pathak et al., 2019, Gupta et al., 2022], since the optimism is being added to the *prior data*, the resulting exploration behaviors differ qualitatively. Adding novelty bonuses to online experience guides the agent to states it has already reached in the online phase, whereas adding optimism to the prior experience encourages the agent to learn to reach new states beyond the regions it has visited through online exploration. Intuitively, while standard optimism leads to exploration along the frontier of online data, optimism on prior data leads to exploration *beyond* the frontier of online data.

**Practical implementation.** In practice, we obtain UCB estimates of the rewards using a combination of a reward model and random-network-distillation (RND) [Burda et al., 2018], a novelty-based reward bonus technique. The former provides a reward estimate of the true reward value and the latter (RND) quantifies the uncertainty of the estimate, allowing us to get a UCB reward estimate.

For the reward model, we update a randomly initialized network $r_\theta(s, a)$ simultaneously with the RL agent on the online replay buffer by minimizing the squared reward loss:

$$\mathcal{L}(\theta) = \mathbb{E}_{(s,a,r) \sim \mathcal{D}} \left[ (r_\theta(s, a) - r)^2 \right].$$

RND randomly initializes two networks $f_\phi(s, a), \bar{f}(s, a)$ that each output an $L$-dimensional feature vector for each state and action. We keep one network, $\bar{f}$, frozen and update the parameters of the

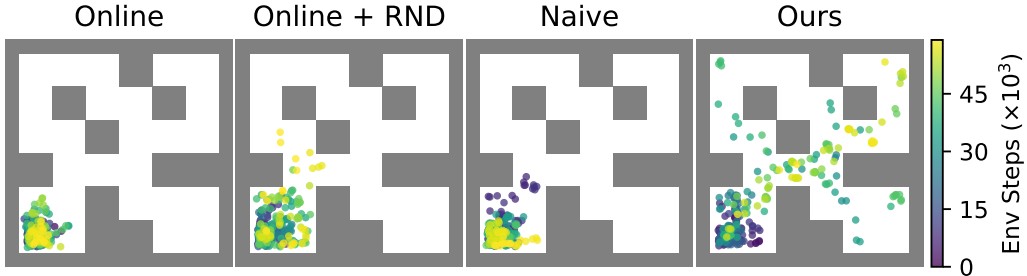

Figure 2: **Visualizations of the agent's exploration behaviors using our optimistic reward labeling strategy on `antmaze-medium`.** The dots shown (500 points each) are sampled uniformly from the online replay buffer (up to 60K environment steps), and colored by the training environment step at which they were added to the buffer. Both **Online** and **Online + RND** do not use prior data, and the latter uses RND reward bonus [Burda et al., 2018] on top of the received online reward to encourage online exploration. **Naïve** learns a reward model and relabels the unlabeled prior data with its reward estimate with no optimism. **Ours** labels the prior data with an optimistic reward estimate.

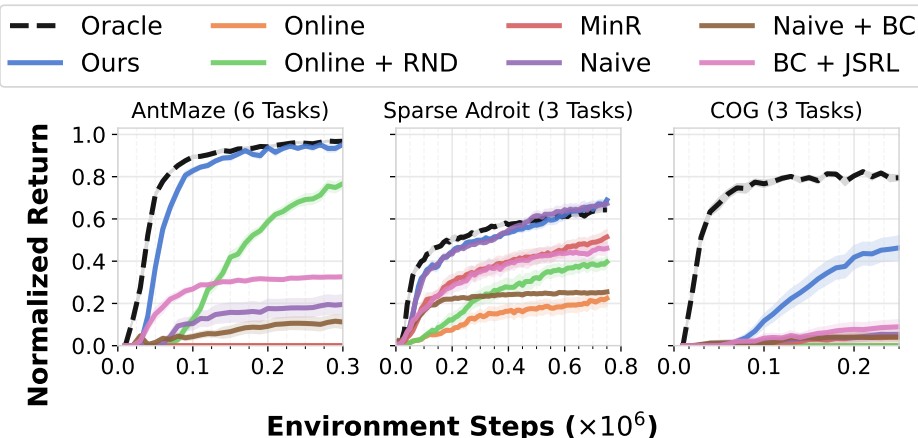

Figure 3: Aggregated results for 6 AntMaze tasks, 3 Adroit tasks and 3 COG tasks. **Ours** is the only method that largely recovers the performance of vanilla RLPD (**Oracle**), which has access to *true* rewards (**Ours** *does not*). The baselines that do not use the prior unlabeled data (**Online** and **Online + RND**) perform poorly on both AntMaze and Adroit. Naïvely labeling the prior data without optimism (**Naïve**) performs competitively on Adroit but poorly on AntMaze. Section 5.2 contains the description for the baselines we compare with.

other network $\phi$ to minimize the mean-square-error between the predicted features and the frozen ones once on every new transition $(s_{\text{new}}, a_{\text{new}}, s'_{\text{new}}, r_{\text{new}})$ encountered:

$$\mathcal{L}(\phi) = \frac{1}{L}\|f_\phi(s_{\text{new}}, a_{\text{new}}) - \bar{f}(s_{\text{new}}, a_{\text{new}})\|_2^2.$$

During online exploration, $\|f_\phi(s,a) - \bar{f}(s,a)\|_2^2$ forms a measure of the agent's uncertainty about $(s, a)$, leading us to the following approximate UCB estimate of the reward function

$$\text{UCBR}(s,a) \leftarrow r_\theta(s,a) + \frac{1}{L}\|f_\phi(s,a) - \bar{f}(s,a)\|_2^2.$$

For the RL agent, we use RLPD [Ball et al., 2023], which is specifically designed for efficient online RL and performs learning updates on both online and labeled offline data. It is worth noting that our method is compatible with potentially any other off-policy RL algorithms that can leverage labeled online and offline data. We experiment with an alternative based on implicit-Q learning in Appendix C.4. Although we only have access to unlabeled data in our setting, we constantly label the prior data with UCB reward estimates, enabling a direct application of RLPD updates.

# 4  Related Work

**Exploration with reward bonuses.** One standard approach for targeted exploration is to add exploration bonuses to the reward signal and train value functions and policies with respect to these optimistic rewards. Exploration bonuses seek to reward novelty, for example using an approximate density model [Bellemare et al., 2016, Tang et al., 2017], curiosity [Pathak et al., 2017, 2019], model error [Stadie et al., 2015, Houthooft et al., 2016, Achiam and Sastry, 2017, Lin and Jabri, 2023], or even prediction error to a randomly initialized function [Burda et al., 2018]. When optimistic rewards are calibrated correctly, the formal counterparts to these algorithms [Strehl and Littman, 2008, Dann et al., 2017] enjoy strong theoretical guarantees for minimizing regret [Azar et al., 2017, Jin et al., 2018]. Prior works typically augment the reward bonuses to the online replay buffer to encourage exploration of novel states, whereas in our work, we add reward bonuses to the offline data, encouraging the RL agent to traverse through trajectories in the prior data to explore more efficiently.

**Using prior data without reward labels.** Prior experience without reward annotations can be used for downstream RL in a variety of ways. One strategy is to use this data to pre-train state representations and extract feature backbones for downstream networks, whether using standard image-level objectives [Srinivas et al., 2020, Xiao et al., 2022, Radosavovic et al., 2022], or those that emphasize the temporal structure of the environment [Yang and Nachum, 2021, Ma et al., 2022, Farebrother et al., 2023, Ghosh et al., 2023]. These approaches are complementary to our method, and in our experiments, we find that optimistic reward labeling using pre-trained representations lead to more coherent exploration and faster learning. Prior data may also inform behavioral priors or skills [Ajay et al., 2020, Tirumala et al., 2020, Pertsch et al., 2021, Nasiriany et al., 2022]; once learned, these learned behaviors can provide a bootstrap for exploration [Uchendu et al., 2022, Li et al., 2023], or to transform the agent's action space to a higher level of abstraction [Pertsch et al., 2020, Singh et al., 2020a]. In addition, if the prior data consists of optimal demonstrations for the downstream task, there exists a rich literature for steering online RL with demonstrations [Schaal, 1996, Vecerik et al., 2017, Nair et al., 2017, Zhu et al., 2020]. Finally, for model-based RL approaches, one in theory may additionally train a dynamics model on prior data and use it to plan and reach novel states [Sekar et al., 2020, Mendonca et al., 2021] for improved exploration, but we are not aware of works that explicitly use models to explore from reward-free offline prior data.

**Using prior data with reward labels.** To train on unlabeled prior data with optimistic reward labels, we use RLPD [Ball et al., 2023], a sample-efficient online RL algorithm designed to train simultaneously on reward-labeled offline data and incoming online experience. RLPD is an offline-to-online algorithm [Xie et al., 2021, Lee et al., 2021, Agarwal et al., 2022, Zheng et al., 2023, Hao et al., 2023, Song et al., 2023] seeking to accelerate online RL using reward-labeled prior experience. While this setting shares many challenges with our unlabeled prior data, algorithms in this setting tend to focus on correcting errors in value function stemming from training on offline data [Nakamoto et al., 2023], and correctly balancing between using highly off-policy transitions from the prior data and new online experience [Lee et al., 2021, Luo et al., 2023]. Our optimistic reward labeling mechanism is largely orthogonal to these axes, since our approach is compatible with any online RL algorithm that can train on prior data.

**Online unsupervised RL.** Online unsupervised RL is an adjacent setting in which agents collect action-free data to accelerate finetuning on some downstream task [Laskin et al., 2021]. While both deal with reward-free learning, they crucially differ in motivation: our setting asks how existing (pre-collected) data can be used to inform downstream exploration for a known task, while unsupervised RL studies how to actively collect new datasets for a (yet to be specified) task [Yarats et al., 2022, Brandfonbrener et al., 2022]. Methods for unsupervised RL seek broad and undirected exploration strategies, for example through information gain [Sekar et al., 2020, Liu and Abbeel, 2021] or state entropy maximization [Lee et al., 2019, Jin et al., 2020, Wang et al., 2020] – instead of the task-relevant exploration that is advantageous in our setting. Successful mechanisms developed for unsupervised RL can inform useful strategies for our setting, since both seek to explore in directed ways, whether undirected as in unsupervised RL or guided by prior data, as in ours.

**Offline meta reinforcement learning.** Offline meta-reinforcement learning [Dorfman et al., 2021, Pong et al., 2021, Lin et al., 2022] also seeks to use prior datasets from a task distribution to accelerate learning in new tasks sampled from the same distribution. These approaches generally tend to focus on the single-shot or few-shot setting, with methods that attempt to (approximately) solve

the Bayesian exploration POMDP [Duan et al., 2016, Zintgraf et al., 2019, Rakelly et al., 2019]. While these methods are efficient when few-shot adaptation is possible, they struggle in scenarios requiring more complex exploration or when the underlying task distribution lacks structure [Ajay et al., 2022, Mandi et al., 2022]. Meta RL methods also rely on prior data containing meaningful, interesting tasks, while our optimistic relabeling strategy is agnostic to the prior task distribution.

## 5  Experiments

Our experiments are aimed at evaluating how effectively our method can improve exploration by leveraging prior data. Through extensive experiments, we will establish that our method is able to leverage the unlabeled prior data to boost online sample efficiency consistently, often matching the sample efficiency of the previous best approach that uses *labeled* prior data (whereas we have no access to labeled rewards). More concretely, we will answer the following questions:

1. *Can we leverage unlabeled prior data to accelerate online learning?*
2. *Can representation learning help obtain better reward labels?*
3. *Does optimistic reward labeling help online learning by improving online exploration?*
4. *How robust is our method in handling different offline data corruptions?*

### 5.1  Experimental Setup

The environments that we evaluate our methods on are all challenging sparse reward tasks, including six D4RL AntMaze tasks [Fu et al., 2020], three sparse-reward Adroit hand manipulation tasks [Nair et al., 2021] following the setup in RLPD [Ball et al., 2023], and three image-based robotic manipulation tasks used by COG [Singh et al., 2020b]. See Appendix B for details about each task. For all three domains, we explicitly remove the reward information from the offline dataset so that the agent must explore to solve the task. We choose these environments as they cover a range of different characteristics of the prior data, where each can present a unique exploration and learning challenge. In Adroit, the prior data consists of a few expert trajectories on the target task and a much larger set of data from a behavior cloning policy. In AntMaze, the prior data consists of goal-directed trajectories with varying start and end locations. In COG, the prior data is mostly demonstration trajectories for sub-tasks of a larger multi-stage task. These tasks are interesting for the following reasons:

**Sparse reward (All).** Due to the sparse reward nature of the these tasks, the agent has no knowledge of the task, it must explore to solve the task successfully. For example, on AntMaze tasks, the goal is for an ant robot to navigate in a maze to reach a fixed goal from a fixed starting point. The prior data for this task consists of sub-optimal trajectories, using starting points and goals potentially different from the current task. Without knowledge of the goal location, the agent must explore each corner of the maze to identify the actual goal location before solving the task is possible.

**High-dimensional image observation (COG).** The three COG tasks use image observations, which poses a much harder exploration challenge compared to state observations because of high dimensionality. As we will show, even with such high-dimensional observation spaces, our algorithm is still able to efficiently explore and accelerate online learning more than the baselines. More importantly, our algorithm can incorporate pre-training representations that further accelerates learning.

**Multi-stage exploration (COG).** The robotic manipulation tasks in COG have two stages, meaning that the robot needs to perform some sub-task (e.g., opening a drawer) before it is able to complete the full task (e.g., picking out a ball from the drawer). See Appendix B, Figure 9 for visualizations of images from the COG dataset showing the two stage structure.

The prior data consists only of trajectories performing individual stages by themselves, and there is no prior data of complete task executions. The agent must learn to explore past the first stage of the task without any reward information, before it can receive the reward for completing the full task. Therefore, this multi-stage experiment requires the agent to be able to effectively make use of incomplete trajectories to aid exploration. This setting is closer to the availability of the data one may encounter in the real world. For example, there may be a lot of data of robots performing tasks that are only partially related to the current task of interest. This data could also be unlabeled, as the rewards for other tasks may not match the current task. Despite being incomplete, we would still like to be able to utilize this data.

## 5.2 Comparisons

While most prior works do not study the setting where the prior data is unlabeled, focusing on either the standard fully labeled data setting or utilizing an online unsupervised learning phase, we can adapt some of these prior works as comparisons in our setting (see Appendix A for details):

**Online**: A data-efficient off-policy RL agent that does *not* make use of the prior data at all.

**Online + RND**: An augmentation of online RL with RND as a novelty bonus. This baseline uses only online data, augmenting the online batch with RND bonus [Burda et al., 2018]. To be clear, of the comparisons listed here in bold, this is the only one that uses an online RND bonus.

**Naïve Reward Labeling**: This comparison labels the prior data reward with an unbiased reward estimate; this is implemented using our method but omitting the RND novelty score.

**Naïve + BC**: This baseline additionally uses a behavioral cloning loss to follow the behaviors as seen in prior data, inspired by similar regularization in offline RL [Vecerik et al., 2017].

**MinR**: This is an adaptation of UDS [Yu et al., 2022] to the online fine-tuning setting. The original UDS method uses unlabeled prior data to improve offline RL on a smaller reward-labeled dataset, using the minimum reward of the task to relabel the unlabeled data. MinR uses the same labeling strategy, but with RLPD for online RL.

**BC + JSRL**: This baseline is an adaptation of JSRL [Uchendu et al., 2022]. The original JSRL method uses offline RL algorithms to pre-train a guide policy using a fully labeled prior dataset. Then, at each online episode, it uses the guide policy to roll out the trajectory up to a random number of steps, then switches to an exploration policy such that the initial state distribution of the exploration policy is shaped by the state visitation of the guide-policy, inducing faster learning of the exploration policy. Since in our setting the prior data has no labels, we use behavior cloning (BC) pre-training to initialize the guide-policy instead of using offline RL algorithms.

**Oracle**: This is an oracle baseline that assumes access to ground truth reward labels, using the same base off-policy RL algorithm [Ball et al., 2023] with true reward labels on prior data.

We also include a behavior prior baseline (inspired by the behavior prior learning line of work [Ajay et al., 2020, Tirumala et al., 2020, Pertsch et al., 2021, Nasiriany et al., 2022, Singh et al., 2020a]) comparison in Appendix C.5 on AntMaze.

## 5.3 Does optimistic labeling of prior data accelerate online learning?

Figure 3 shows the aggregated performance of our approach on the two state-based domains and one image-based domain. On AntMaze and COG domains, our optimistic reward labeling is able to outperform naïve reward labeling significantly, highlighting its effectiveness in leveraging the prior data to accelerate online learning. Interestingly, in the two state-based domains, we show that without access to prior rewards, our method can nearly match the performance of the oracle baseline with prior rewards. This suggests that optimistic reward labeling can allow RL to utilize unlabeled prior data almost as effectively as labeled prior data on these three domains. In particular, on the sparse Adroit `relocate` task, using optimistic rewards can even outpace the oracle baseline (Appendix C.2, Figure 13). In the COG domain, there is a larger performance gap between our method and the oracle. We hypothesize that this is due to the high-dimensional image observation space, causing the exploration problem to be more challenging. On the Adroit domain, we find that periodic resetting of the learned reward function is necessary to avoid overfitting to the collected online experience – with the resetting, we find that even naïve reward relabeling results in exploratory behavior, diminishing the gap when performing explicit reward uncertainty estimation with RND.

## 5.4 Can representation learning help obtain better reward labels?

We now investigate whether we can improve exploration by basing our UCB reward estimates on top of pre-trained representations acquired from the prior data. We chose to use representations from ICVF [Ghosh et al., 2023], a method that trains feature representations by pre-training general successor value functions on the offline data. Value-aware pre-training is a natural choice for downstream reward uncertainty quantification, because the pre-training encourages representations to obey the spatiality of the environment – that states close to one another have similar representa-

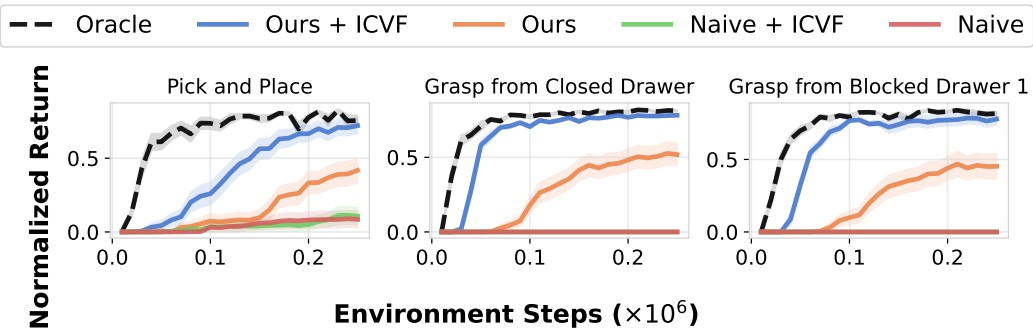

Figure 4: **Accelerating exploration with pre-trained representations on three visual-input COG tasks.** **Ours + ICVF** uses optimistic reward labeling with a pre-trained representation initialization; **Ours** uses optimistic reward labeling without the pre-trained representation initialization. The same applies for **Naïve** and **Naïve + ICVF**. Overall, initializing the reward model and the RND network using pre-trained representations greatly increases how quickly the model learns.

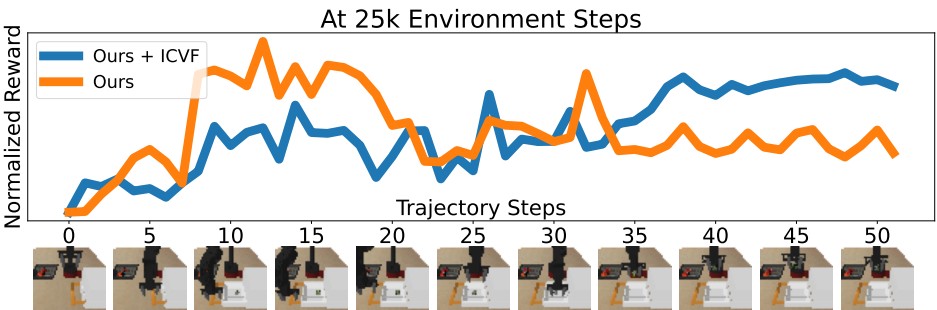

Figure 5: **The effect of pre-trained representations on the relabeled reward**. We show the normalized relabeled rewards (by standardizing it to have zero mean and unit variance) of an optimal prior trajectory from the closed drawer task in the COG environment. Using pre-trained representations to learn the optimistic rewards leads to smoother rewards over the course of an optimal trajectory. See more examples in Appendix D.

tions, while states that are difficult to reach from one another are far away. We pre-train an ICVF model on the prior dataset and extract the visual encoder $\xi_{\text{ICVF}}(s)$, which encodes an image into a low-dimensional feature vector. These pre-trained encoder parameters are used to initialize the encoder for both the reward model $r_\theta(\xi_\theta(s))$ and the RND network $f_\phi(\xi_\phi(s))$. Note that these encoder parameters are not shared and not frozen, meaning that once training begins, the encoder parameters for the reward model will be different from the encoder parameters for the RND model, and also diverge from the pre-trained initialization. As shown in Figure 4, ICVF consistently improves the online learning efficiency across all image-input tasks on top of optimistic reward labeling. It is worth noting that the encoder initialization from pre-training alone is not sufficient for the learning agent to succeed online (Naïve + ICVF still fails almost everywhere). On most tasks, without optimistic reward labeling, the agent simply makes zero progress, further highlighting the effectiveness of optimistic labeling in aiding online exploration and learning. To understand how the pre-trained representation initialization influences online exploration, we analyze its effect on the UCB reward estimate early in the training. Figure 5 shows the UCB reward estimates on an expert trajectory of the `Grasp from Closed Drawer` task when the reward model and the RND model are initialized with the pre-trained representations (in blue) and without the pre-trained representations (in orange). When our algorithm is initialized with the ICVF representations, the UCB reward estimates are increasing along the trajectory time step, labeling the images in later part of the trajectory with high reward, forming a natural curriculum for the agent to explore towards the trajectory end. When our algorithm is initialized randomly without the pre-trained representations, such a clear monotonic trend disappears. We hypothesize that this monotonic trend of the reward labeling may account for the online sample efficiency improvement with a better shaped reward.

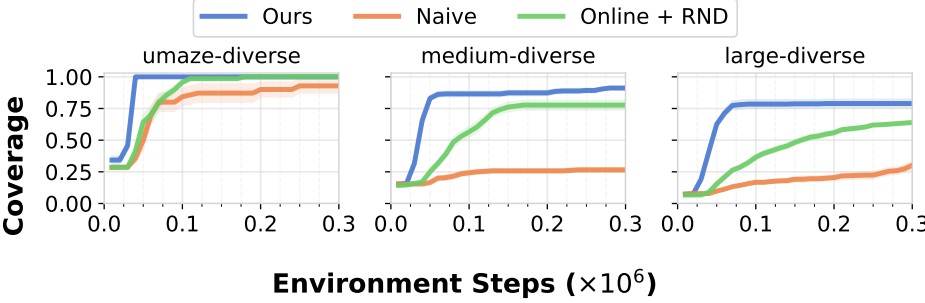

Figure 6: **State coverage for 3 AntMaze tasks**, estimated by counting the number of square regions that the ant has visited, normalized by the total number of square regions that can be visited. **Ours** optimistically labels the prior data, whereas **Naïve** naïvely labels the prior data without optimism. Optimistic labeling enables the RL agent to achieve a significantly higher coverage. **Online RND** also explores well in the maze, but with a much slower rate compared to our approach, highlighting the importance of leveraging unlabeled prior data for efficient online exploration. Full results in Appendix C.1.

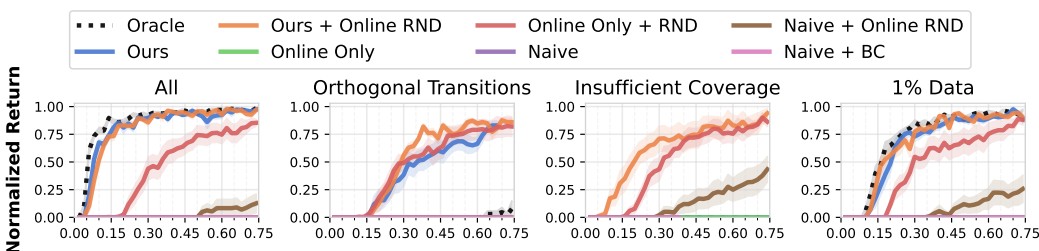

Figure 7: The normalized return on `antmaze-large-diverse-v2` under different offline data corruptions (see visualization of the offline data for each kind in Appendix B, Figure 8).

## 5.5 Does optimistic reward labeling help online learning by improving online exploration?

We have shown that optimistic reward labeling is effective in accelerating online learning, does it actually work because of better exploration? To answer this question, we evaluate the effect of our method on the state-coverage of the agent in the AntMaze domain. In this domain, the agent is required to reach a particular goal in the maze but has no prior knowledge of the goal, and thus required to explore different parts of the maze in the online phase in order to locate the goal. The optimal behavior that we expect the agent to perform is to quickly try out every possible part of the maze until the goal is located, and then focus entirely on that goal to achieve 100% success rate.

To more quantitatively measure the quality of our exploration behavior, we divide up the 2-D maze into a grid of square regions, and count the number of regions that the Ant agent has visited (using the online replay buffer). This provides a measure for the amount of state coverage that the agent has achieved in the environment, and the agent that explores better should achieve higher state coverage. Figure 6 shows how optimistic reward labeling affects the state coverage, and Figure 2 shows an example visualization of the state-visitation for each method on `antmaze-medium`. Across all six AntMaze tasks, our approach achieves significantly higher state-coverage compared to naïvely label without optimism both in the early stage of training and asymptotically compared to the baseline that does not leverage the unlabeled prior data. In particular, we find that naïvely labeling reward with no optimism is doing poorly on larger mazes, highlighting the importance of optimistic reward labeling for effective online exploration.

## 5.6 How robust is our method in handling different offline data corruptions?

To further test the capability of our method in leveraging prior data, we corrupt one of the D4RL offline datasets, `antmaze-large-diverse-v2`, and test how our method performs under the corruption. We experimented with three different corruptions (and reported results in Figure 7).

**Orthogonal transitions:** all the transitions that move in the up/right (towards the goal) are removed from the offline data. In this setting, stitching the transitions in the offline data would not lead to a trajectory that goes from the start to the goal. This is a good test for the robustness of the algorithm for exploration because the agent can not rely on the actions in the prior data and must explore to discover actions that move the ant in the correct direction.

**Insufficient coverage:** all the transitions around the goals are removed from the offline data. This requires the algorithm to actively explore so that it discovers the desired goal that it needs to reach.

**1% data:** A random 1% of the transitions are kept in the offline data. This tests the capability of our method in handling the limited data regime.

In addition to the main comparisons above, we consider two modifications specifically for these experiments to address the more challenging tasks.

**Ours + Online RND**: This is an extension of our method, with an RND bonus added to the online data in addition to the prior data. Similar to how we can extend our method with representation learning, having exploration bonuses on the online data does not conflict with exploration bonuses added to the offline data.

**Naïve + Online RND**: This is an extension of Naïve Reward Labeling. An RND bonus added to the online data, and the prior data is labeled with the unbiased reward estimate as before.

Directly using our method in the *Insufficient Coverage* setting yields zero success, but this is to be expected as our algorithm does not incentivize exploration beyond the offline data. We find that this can be patched by simply combining our method with an additional RND bonus on online data, allowing our method to learn even when data has poor coverage and no data near the goal is unavailable. Similarly, we have found that combining our method with online RND can also boost the performance in the *Orthogonal Transitions*, where no transitions move in the direction of the goal. For the *1% Data* setting, our method can still largely match the oracle performance without the need for an online RND bonus.

It is also worth noting that in the *Orthogonal Transitions* setting, our method achieves good performance whereas the oracle fails to succeed. The failures of the oracle are not unexpected because the prior data do not contain any single trajectories, or even trajectories stitched together from different transitions, that can reach the goal. However, since our method succeeds, this indicates that exploration bonuses applied only to the prior data can still be utilized, despite there being no forward connection through the prior data. These additional results, while initially surprising even to us (especially the "orthogonal" setting, which seems very difficult), strongly suggest that our method can lead to significant improvement even when the data is not very good – certainly such data would be woefully inadequate for regular offline RL or naïve policy initialization.

## 6  Discussion

We showed how we can effectively leverage unlabeled prior data to improve online exploration by running a standard off-policy RL algorithm on the data relabeled with UCB reward estimates. In practice, the UCB estimates can be approximated by combining the prediction of a reward model and an RND network, both can be trained online with little additional computational cost. We demonstrated the surprising effectiveness of this simple approach on a diverse set of domains. Our instantiation of the optimistic reward labeling idea presents a number of avenues for future research. First, on the `relocate` Adroit task, we found that naïvely fitting the reward model on the online replay buffer without any regularization leads to poor performance (possibly due to catastrophic overfitting). While we have found a temporary workaround by periodically re-initializing the reward model, such a solution may seem ad-hoc and could be disruptive to the RL learning due to the sudden change in the learning objective (which is shaped by the reward model). Next, the UCB estimates of the state-action can rapidly change, especially in the beginning of the training, which may cause learning instability in the RL algorithm. There is no mechanism in our current algorithm to tackle such a potential issue. Nevertheless, our work indicates that there exist simple ways of incorporating prior data for online RL agents even when no reward labels exist. Scaling these mechanisms to more complex prior datasets is an exciting direction towards the promise of RL agents that can leverage general prior data.

**Acknowledgement.** This work was partially done while QL was a visiting student researcher at FAIR, Meta. We would like to thank Seohong Park for providing his implementation of goal-conditioned IQL (GC-IQL) and pre-trained GC-IQL checkpoints (which were used in producing Figure 17). We would like to thank Fangchen Liu for the acronym of the method. We would also like to thank Laura Smith, Toru Lin, Oleg Rybkin, Kuba Grudzien, Aviral Kumar, and Joey Hong for discussion on the method and feedback on the early draft of the paper. We would also like to thank the members of the RAIL lab for insightful discussions on the paper. This research was partially supported by the Office of Naval Research under N00014-21-1-2838 and N00014-22-1-2773, and ARO W911NF-21-1-0097.

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

# A    Experiment Setup Details

Our codebase is based on the official RLPD codebase `https://github.com/ikostrikov/rlpd` with minor modifications. All configurations in our experiments use 10 seeds (20 seeds for all COG experiments) and we compute the standard error to plot the confidence interval in all our plots.

## A.1    Architecture Details

RLPD extends a standard off-policy actor-critic algorithm which trains an actor network and a critic network simultaneously. Our extension of the RLPD requires training three additional networks: a reward prediction network, a termination prediction network and an RND network. For state-based domains, each of the actor, critic, reward and RND network is multilayer perceptron (MLP) with ReLU [Agarap, 2018] activation. For the COG (image-based) domain, each network uses a pixel encoder followed by a MLP.

**Pixel encoder architecture.** For the environments in the COG domain, we use the following pixel encoder architecture, based on the pixel encoder implemented in RLPD's official codebase (located at `https://github.com/ikostrikov/rlpd/blob/main/rlpd/networks/encoders/d4pg_encoder.py`). Changes in orange are made to the observation size and number of frames stacked to match the dataset from the COG domain. Following the implementation from Ball et al. [2023], between the actor and the critic, the pixel encoder parameters are shared, and only the gradients through the critic loss objective update the encoder. The RND model, reward model, and the termination model each has its own pixel encoder with no parameter sharing.

| Parameter | Value |
|---|:---:|
| Random Crop Padding | 4 |
| Number of Stacked Frames | 1 |
| Observation Size | [48, 48, 3] |
| CNN Features | [32, 64, 128, 256] |
| CNN Filter Size | 3 |
| CNN Padding | Valid |
| CNN Stride | 2 |
| Encoder Output Latent Dimension | 50 |

Table 1: Pixel Encoder Architecture

**MLP architecture.** In the state-based domains (AntMaze and Sparse Adroit), we use a 3-layer MLP with hidden dimensions of 256. In the COG domain, we use a 2-layer MLP with hidden dimensions of 256, after a pixel encoder (Table 1). The RND network outputs a feature size of $L = 256$. The reward network output a single scalar. For the critic network, layer norm [Ba et al., 2016] is being added to all the layers (of the MLP) with the learnable bias and scaling except the last layer.

**Action parameterization.** In all of our experiments, we follow RLPD's actor design where the actor outputs a mean value $\mu_i$ and a log standard deviation value $\log \sigma_i$ for each action dimension, leading to a diagonal Gaussian distribution, $N(\boldsymbol{\mu}, \text{diag}(\boldsymbol{\sigma}))$. This distribution is then transformed through the Tanh function such that the range of the action falls in $(-1, 1)$. The log standard deviation is clipped to be between $-20$ and $2$.

**Termination prediction network.** While predicting termination is trivial for the COG domain and the Sparse Adroit domain (the episode never terminates before the maximum episode length), predicting termination in AntMaze is just as difficult as predicting the reward (e.g., the episode terminates). To make our implementation generally applicable to any environments with or without terminations, we train a termination prediction network which we briefly mentioned in the beginning of the section. More concretely, the termination prediction network also takes in a state-action pair and outputs a scalar, which predicts the probability of the episode terminating at the current transition through a Sigmoid function. Using the termination prediction network, the termination is also being labeled in the prior unlabeled data. This results in a slightly different TD back-up equation, which

can be easily integrated in the RLPD algorithm:

$$Q(s, a) \leftarrow \text{UCBR}(s, a) + \gamma(1 - \hat{T}(s, a))\bar{Q}(s', a'),$$

where $\hat{r}(s, a)$ is the labeled reward value and $\hat{T}(s, a)$ is the labeled termination prediction (with 1 meaning termination with a probability of 1). We left the discussion of the termination prediction network out of the main body of the paper because it is a minor implementation detail. For completeness, we present the more detailed version of Algorithm 1 with the addition of the termination prediction as Algorithm 2.

**Training of the UCB networks.** The training of the reward prediction model, termination prediction model and the RND model starts after 10000 environment steps for the AntMaze and the Sparse Adroit domains and 5000 environment steps for the COG domain. The RND network takes 1 gradient step on the current transition $(s_{\text{new}}, a_{\text{new}})$ according to the following objective:

$$\mathcal{L}(\phi) = \frac{1}{L}\|f_\phi(s_{\text{new}}, a_{\text{new}}) - \bar{f}(s_{\text{new}}, a_{\text{new}})\|_2^2,$$

where $L$ is the number of output features of the RND network which is set to be 256. Both the reward model and the termination prediction model take the same number of gradient steps as the actor/critic (1 for COG and 20 for AntMaze and Sparse Adroit). The termination prediction model optimizes the following objective (the transition $(s, a, r, T)$ is sampled from the online replay buffer $\mathcal{D}$) with $T \in \{0, 1\}$ being the binary variable that indicates whether the episode terminates at the current transition:

$$\mathcal{L}(\theta_{\text{term}}) = -\mathbb{E}_{(s,a,T)\sim\mathcal{D}}\left[T \cdot \text{Log\_Sigmoid}(\hat{T}_{\theta_{\text{term}}}(s, a)) + (1 - T) \cdot \text{Log\_Sigmoid}(-\hat{T}_{\theta_{\text{term}}}(s, a)))\right],$$

and the reward prediction model optimizes the following objective:

$$\mathcal{L}(\theta) = \mathbb{E}_{(s,a,r)\sim\mathcal{D}}\left[(r_\theta(s, a) - r)^2\right].$$

---

**Algorithm 2** EXPLORE (with termination estimates)

---

1: **Input:** prior unlabeled data $\mathcal{D}_{\text{offline}}$
2: Initialize the UCB estimate of the reward function: $\text{UCBR}(s, a)$
3: Initialize an estimate of the termination function: $\hat{T}(s, a)$
4: Online replay buffer $\mathcal{D} \leftarrow \emptyset$
5: Initialize off-policy RL algorithm with a policy $\pi$.
6: **for** each environment step **do**
7:     Collect $(s, a, s', r)$ using the policy $\pi$ and add it to the replay buffer $\mathcal{D}$.
8:     Update the UCB estimate $\text{UCBR}(s, a)$ using the new transition $(s, a, s', r)$.
9:     Relabel each transition $(s, a, s')$ in $\mathcal{D}_{\text{offline}}$ with $\hat{r} = \text{UCBR}(s, a), \hat{T} = \hat{T}(s, a)$.
10:     Run off-policy RL update on both $\mathcal{D}$ and the relabeled $\mathcal{D}_{\text{offline}}$ to improve the policy $\pi$.
11: **end for**

---

**Modifications to UCB networks for the COG domain**. For the COG domain, we used ICVFs to test the incorporation of representation learning methods. We wanted to experiment with the formulation where the UCB network is learned on top of the encoder outputs of the ICVF representations, which depend only on states/observations $s$. That is, for some observation $s$ and encoder $\xi(s)$, we wished to learn a UCB reward model $r(\xi(s))$. However, this formulation does not incorporate actions. Therefore, for all our experiments on the COG domain, the RND network, reward model, and termination prediction network all operate on observations only. More concretely, we make the following replacements to our network architecture for the COG domains: for the RND networks we replaced $f_\phi(s, a)$ with $f_\phi(s)$ and $\bar{f}(s, a)$ with $\bar{f}(s)$, for the reward model, we replaced $r_\theta(s, a)$ with $r_\theta(s)$, and for the termination prediction model, we replaced $\hat{T}(s, a)$ with $\hat{T}(s)$.

## A.2 RLPD Hyperparameters

Most of the hyperparameters we used are unchanged from the original RLPD hyperparameters, listed below.

| Parameter | Value |
|---|---|
| Online batch size | 128 |
| Offline batch size | 128 |
| Discount factor $\gamma$ | 0.99 |
| Optimizer | Adam |
| Learning rate | $3 \times 10^{-4}$ |
| Critic ensemble size | 10 |
| Random critic target subset size | 2 for Adroit and COG and 1 for AntMaze |
| Gradient Steps to Online Data Ratio (UTD) | 20 |
| Network Width | 256 |
| Initial Entropy Temperature | 1.0 |
| Target Entropy | $-\dim(\mathcal{A})/2$ |
| Entropy Backups | False |
| Start Training | after 5000 environment steps |

Table 2: RLPD hyperparameters.

The online and offline batch sizes change when we run baselines that do not use offline data. In that case, the offline batch size becomes 0, and the online batch size becomes 256, to keep the combined batch size the same. The update the data ratio (UTD) was set to 1 in the COG domain. For the other domains, we used the default value of 20. Action repeat is a hyperparameter that determines the number of times an action sampled from the policy is repeated in the environment. It was originally 2 for the pixel-based hyperparameters in RLPD. We set it to 1 to match the prior dataset from the COG domains.

### A.3  Baseline Hyperparameters and Tuning Procedure Description

**Online.** For the online baseline, we don't use any offline data. So the offline batch size is 0 and the online batch size is 256.

**Online + RND.** For the online with RND baseline, we added an RND bonus on top of the ground truth rewards in the online data. That is, given an online transition $(s, a, r, s')$, and RND feature networks $f_\phi(s, a), \bar{f}(s, a)$, we set

$$\hat{r}(s, a) \leftarrow r + \frac{1}{L} ||f_\phi(s, a) - \bar{f}(s, a)||_2^2$$

and use the transition $(s, a, \hat{r}, s')$ in the online update. The RND training is done the same way as in our method where a gradient step is taken on every new transition collected.

**Naïve Reward Labeling.** The naïve reward labeling baseline is a simple way to incorporate the unlabeled prior data by learning a reward model online and relabeling the rewards of the prior data. For some prior transition $(s, a, s')$, and a reward model $r_\theta$ learned from online interactions, we set the reward of this prior transition to be $\hat{r}(s, a) = r_\theta(s, a)$, giving the relabeled offline transition $(s, a, \hat{r}, s')$. Then, we take a batch of 50% offline data and 50% online data, and run RLPD as usual. Note that the only difference this baseline has with our method is that $\hat{r}$ does not have an RND bonus added to it.

**Naïve + BC.** Aside from performing the reward relabeling naïvely as described in the previous section, we also regularize the actor to more closely match the behavior in the prior data by adding a behavior cloning loss to the actor loss ($\theta_{\text{actor}}$ is the parameter of the actor in RLPD):

$$\mathcal{L}_{\text{actor,bc}}(\theta_{\text{actor}}) = \mathcal{L}_{\text{actor}}(\theta_{\text{actor}}) - \alpha_{\text{bc}}\mathbb{E}_{(s,a)\sim\mathcal{D}_{\text{offline}}} \left[\log \pi_{\theta_{\text{actor}}}(a|s)\right],$$

where $\mathcal{L}_{\text{actor}}$ is the original loss function used in RLPD.

To find the best loss coefficient $\alpha_{\text{bc}}$, we perform a sweep over $\{0., 0.01, 0.1\}$ for each domain. The best coefficient is 0.01 for all the domains.

**MinR.** The MinR baseline is a simpler version of Naïve Reward Labeling. For all prior transitions $(s, a, s')$, we set the reward $\hat{r}(s, a) = r_{\text{min}}$, where $r_{\text{min}}$ is the minimum reward for the task. For example, for sparse 0-1 rewards, $r_{\text{min}} = 0$. Then, we run RLPD with this relabeled offline data.

**BC + JSRL.** This baseline has a few additional hyperparameters. One controls the number of gradient steps to pre-train the behavior cloning guide policy on the unlabeled prior data, $N_{\text{JSRL}}$. Each gradient step is taken according to the BC objective ($\theta_{\text{guide\_policy}}$ is the parameters for the guide policy):

$$\mathcal{L}_{\text{JSRL,bc}}(\theta_{\text{guide\_policy}}) = -\mathbb{E}_{(s,a) \sim \mathcal{D}_{\text{offline}}} \left[ \log \pi_{\theta_{\text{guide\_policy}}}(a|s) \right].$$

The other controls the probability of starting a trajectory by first rolling out the guide policy, $\beta_{\text{JSRL}}$. Finally, in the course of a trajectory, we switch from the guide policy to the exploration policy with probability $1 - \gamma$ where $\gamma = 0.99$ is the discount factor. To find the best hyper-parameters, we perform a sweep over $\beta_{\text{JSRL}} \in \{0.1, 0.5, 0.9\}$ and $N_{\text{JSRL}} \in \{5000, 20000, 100000\}$ for both AntMaze and Sparse Adroit. For AntMaze, the best parameter set is $(\beta_{\text{JSRL}}, N_{\text{JSRL}}) = (0.9, 5000)$. For Sparse Adroit, the best parmaeter set is $(\beta_{\text{JSRL}}, N_{\text{JSRL}}) = (0.5, 100000)$. For COG domain, none of the sweep runs succeeded. We just report the results of a randomly picked parameter set: $(\beta_{\text{JSRL}}, N_{\text{JSRL}}) = (0.5, 100000)$. All the return evaluations are done solely on the exploration policy (without rolling out the guide policy).

**Oracle.** This baseline assumes that the ground truth rewards for the online task are given in the offline data, and runs RLPD with that data.

**ICVF training details.** To evaluate reward functions and uncertainty estimation, we run our method on top of a ICVF representation [Ghosh et al., 2023] pre-trained on the offline prior data. We use the open-source implementation from the authors at `https://github.com/dibyaghosh/icvf_release`, using the COG encoder as the visual backbone and the default hyperparameters otherwise from the open-source implementation. We pre-train the model for $0.75 \times 10^5$ timesteps, and use the final checkpoint as the encoder initialization for both RND and the reward model predictor. Notably, the representation is only ever used as an initialization, and is not frozen or regularized during reward training.

| Parameter | Value |
|---|---|
| Batch size | 256 |
| Discount factor $\gamma$ | 0.99 |
| Optimizer | Adam |
| Learning rate | $3 \times 10^{-4}$ |
| MLP Width | 256 |
| MLP layers | 2 |

Table 3: ICVF hyperparameters.

# B   Domain Details

**AntMaze (state)**   In the D4RL AntMaze Fu et al. [2020] environment, an 8-DoF quadrupedal "ant" agent navigates a maze from a fixed starting location and towards a fixed goal location. The reward is sparse: 0 around the goal and -1 elsewhere. The observation space is 29 dimensions, with the first 2 dimensions indicating the position of the ant within the maze. The action space is 8 dimensional. Trajectories terminate upon reaching the goal. We tested 6 total AntMaze tasks within a total of 3 mazes of increasing size and complexity: `umaze`, `umaze-diverse`, `medium-play`, `medium-diverse`, `large-play`, and `large-diverse`.

The offline data for this environment consists of trajectories with many different start and end positions. The nature of these positions depends on the type of environment. In the `play` AntMaze environments, the start and end positions are handpicked whereas in the `diverse` AntMaze environments, they are randomly chosen. Since the actual task contains only a single fixed start and goal, the offline data does not always correspond to the online task. The coverage of the offline data is relatively complete over the entire maze.

**Adroit (state)**   The Adroit environment Nair et al. [2021] we used consisted of a 24-DoF hand robot agent and 3 dexterous manipulation tasks: pen-spinning, door-opening, and ball relocation. The data distribution for the Adroit environments is relatively narrow. The prior data for these environments consists of expert trajectories from human demonstrations, as well as many behavior-cloned trajectories on the expert data. As the prior data here is closer to the target task distribution

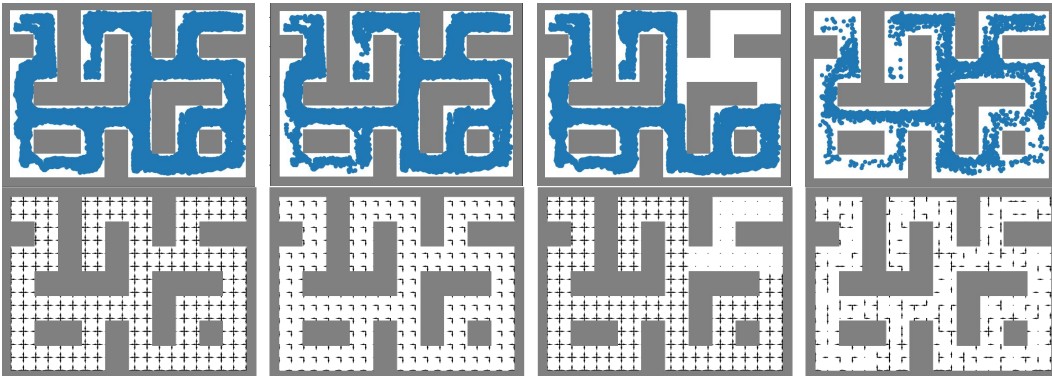

Figure 8: Additional visualizations on offline dataset with four different kinds of corruptions. **Top**: visualization of the coverage of the transitions in the offline data. **Bottom**: visualization of the direction of the transitions in the offline data. **Left**: All, **Middle Left**: Orthogonal Transitions, **Middle Right**: Insufficient Coverage, **Right**: 1% Data.

compared to other environments, these experiments in Adroit allow us to compare the performance of our method more critically against our baselines that are less exploration focused.

**COG (vision)** The environments and datasets from COG come from the repo: `https://github.com/avisingh599/cog`. In the COG environments Singh et al. [2020b], the actor is a simulated WidowX robot arm. The observation space is image-based with dimensions $(48, 48, 3)$. The rewards are sparse, with +1 reward when the task is completed, and 0 reward otherwise. The horizon is infinite, but for exploration and evaluation, we set a max path length to end the episode.

The first environment we call `Pick and Place`. For this environment, the arm is placed in front of a small object and tray. The task is to pick up / grasp the object and place it in the tray, where success occurs when the object is in the tray. The object is initialized at different locations on the ground. The maximum episode length is 40.

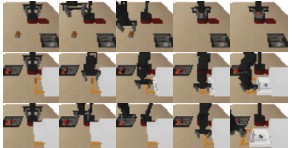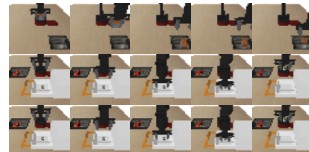

Figure 9: **Visualizations of data used in the COG tasks**. The left block contains the first stage of the task, and the right block contains the second stage of the task. The first row is for the `Pick and Place` task, the second row is picking out a ball from a closed drawer, the `Grasp from Closed Drawer` task, and the third row is picking out a ball from a blocked drawer, the `Grasp from Blocked Drawer 1` task.

The offline dataset for the `Pick and Place` consists of trajectories of grasping attempts and separate trajectories of placing attempts. The data for these environments comes from a scripted policy that achieves roughly 40% success at grasping the object and roughly 90% success at placing the object into the tray. There are 100 trajectories of grasping attempts and 50 trajectories of placing attempts. There may be overlaps in states in the grasping and placing data, but there is no trajectory that completes the entire pick and place task. For our RLPD baseline, where the rewards from the prior dataset are considered, we set a reward of +1 when the object is successfully placed in the tray, and 0 for all other transitions.

We ran experiments on two drawer environments from COG. The main interaction in these environments is between the WidowX robot arm and a two tiered drawer. The first setup we call `Grasp from Closed Drawer`. In this setup, the bottom drawer is closed, and an object is inside the bottom drawer. The task is to open the bottom drawer and pick out the object. The second setup we call `Grasp from Blocked Drawer 1`. In this setup, the bottom drawer is closed and blocked by an open upper drawer. The task is to close the upper drawer, open the bottom drawer, and pick out the object. For both drawer environments, a success occurs when the object is taken out of the

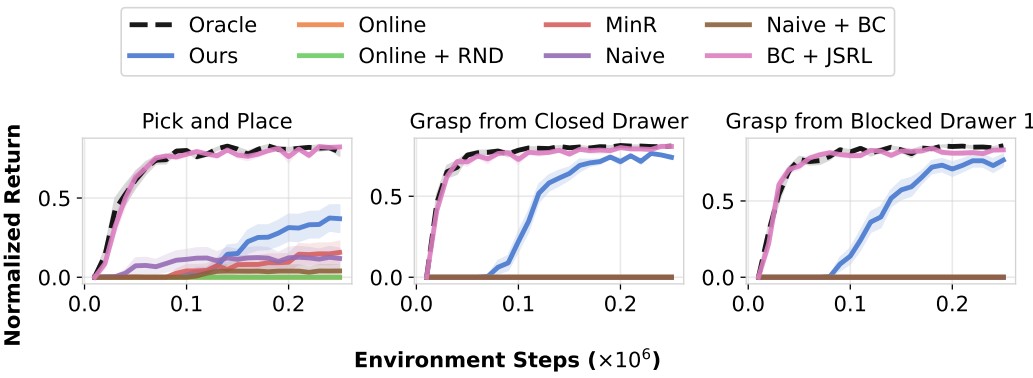

Figure 10: The performance on 3 COG tasks with a larger offline dataset. **BC+JSRL** is able to match the oracle performance. In the main body of the paper (Figure 3), we are using a dataset that is roughly 1% in size.

bottom drawer. The maximum episode length is 50 for `Grasp from Closed Drawer`, and 80 for `Grasp from Blocked Drawer 1`.

The offline dataset we used for the drawer tasks consists of trajectories of drawer opening attempts and separate trajectories of taking the object out of an open drawer. For example, in the `Grasp from Closed Drawer` task, the drawer opening attempts consist of trajectories of opening the bottom drawer. Meanwhile, for the `Grasp from Blocked Drawer 1` task, the drawer opening attempts consist of trajectories of closing the top drawer and opening the bottom drawer. For both tasks, the trajectories of taking the object out of an open drawer are the same. Similar to `Pick and Place`, these trajectories were generated using a randomized scripted policy as described in Singh et al. [2020b] that has about a 40-50% success rate at opening the drawer and about a 70% success rate at taking the object out of the drawer. Like the `Pick and Place` task, there is no trajectory that completes the entire task, but there may be overlaps in states. For our RLPD baseline, we set a reward of +1 when the object has been successfully taken out of the bottom drawer, and 0 for all other transitions. There are 25 trajectories of drawer opening attempts and 25 trajectories of object-taking attempts for the two drawer tasks.

Initially, we experimented with a larger offline dataset with 100 times more trajectories for these three tasks and we have found that the **BC+JSRL** baseline to closely match the oracle performance and our method did not really an edge over that baseline (see Figure 10). This is perhaps to be expected as **BC+JSRL** puts a strong prior on following the offline data action by behavior cloning, and when offline data are good demonstration data. On the other hand, our method does not assume that the offline dataset is of good quality, and does not try to closely follow the actions. Consequently, our method is able to perform well more robustly across different settings compared to **BC+JSRL** as shown in our main results (Figure 3).

## C    Additional Results

### C.1    Full AntMaze results.

We include the complete set of AntMaze results in this section. Figure 11 shows the normalized return, which is the percentage success rate of the ant reaching the desired goal. Figure 12 shows the coverage metric, which is an estimate of how much percentage of the free space that the ant has explored in the maze.

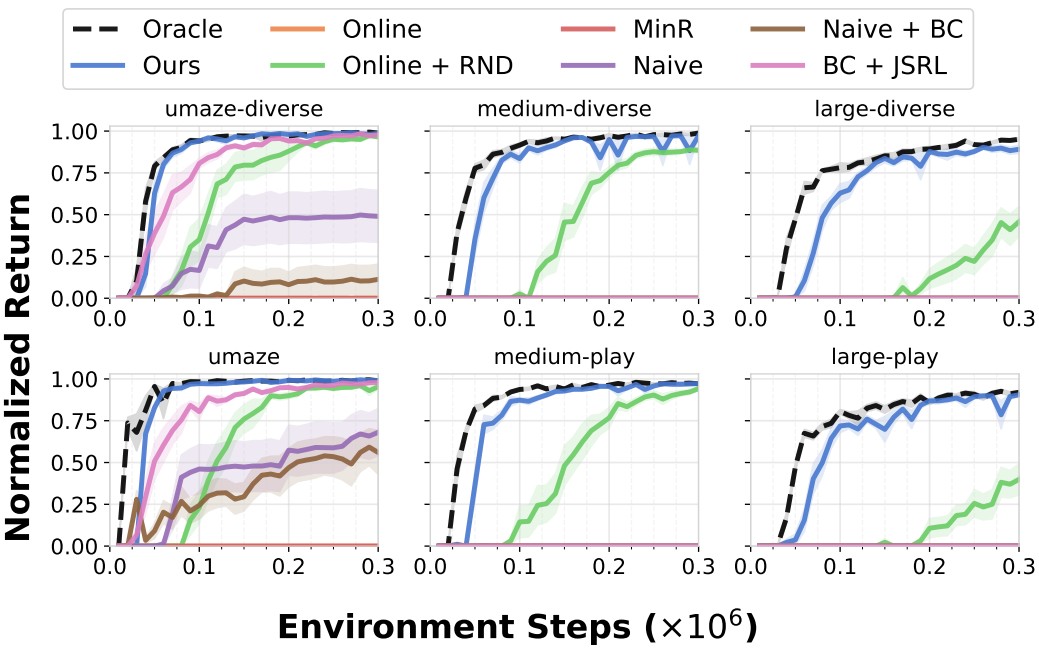

Figure 11: **The success rate for the 6 AntMaze tasks.** Every method except **Oracle** has no access to the reward function. Our method consistently performs well and nearly matches the oracle performance.

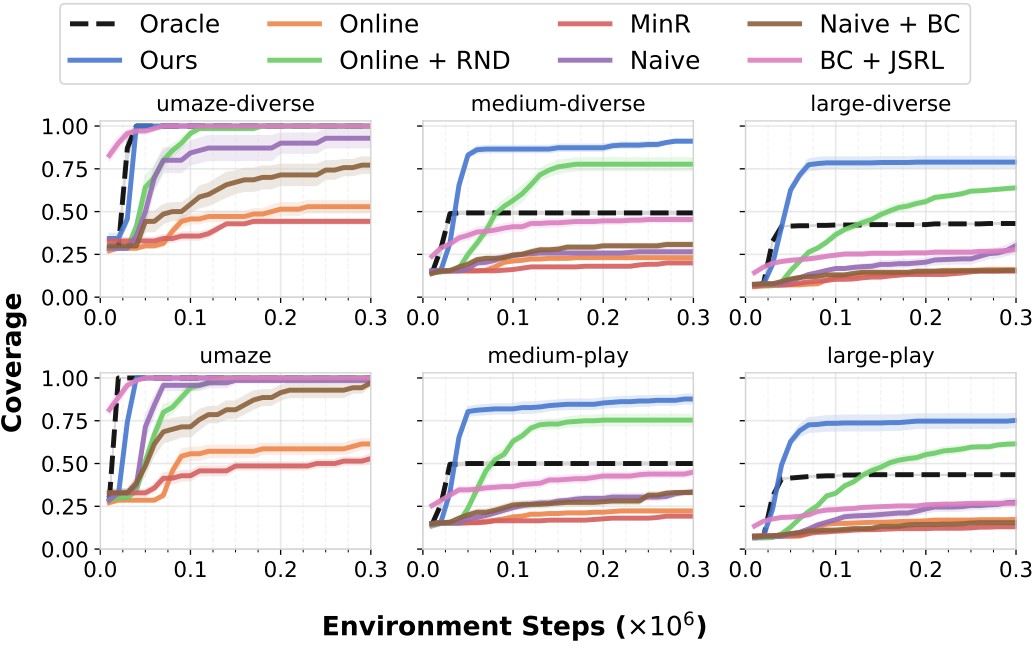

Figure 12: **State coverage for the 6 AntMaze tasks**, estimated by counting the number of square regions that the ant has visited, normalized by the total number of square regions that can be visited.

## C.2 Full Adroit Results

We include the complete set of Adroit results in this section. Figure 13 shows the normalized return for each method on each of the three sparse Adroit tasks. Figure 14 studies the effect of periodic resetting the reward and the RND networks on the hardest task, `relocate`.

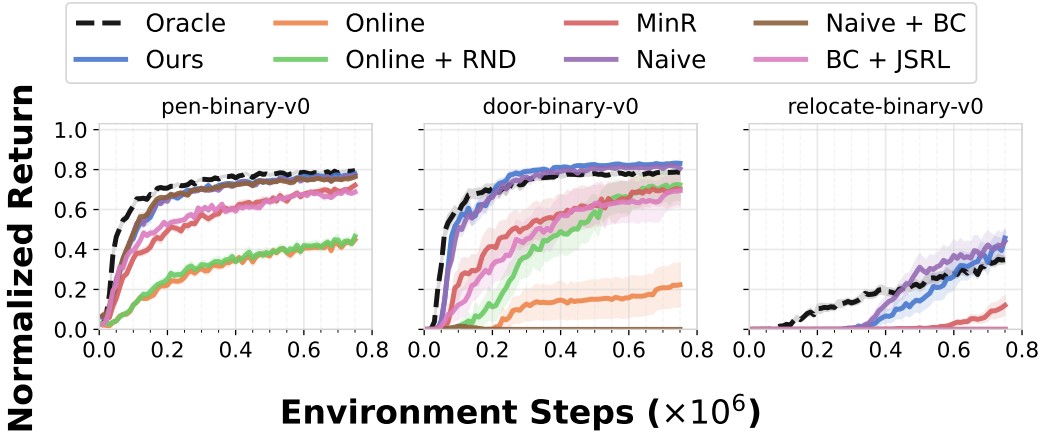

Figure 13: **The normalized return for the 3 sparse Adroit tasks.** Every method except **Oracle** has no access to the reward function. Naïve reward modeling without optimistic relabeling is sufficient for all Adroit tasks. Periodically resets (every 1000 environment steps) are used for the `relocate` task. See Figure 14 below for the reset ablation study.

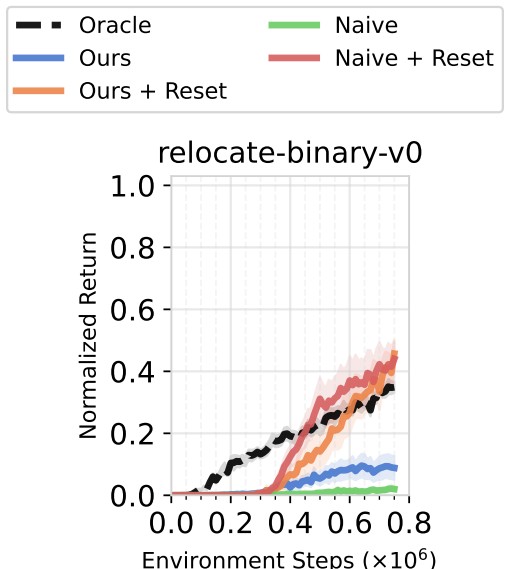

Figure 14: **Periodically resetting the reward model is crucial for success in `relocate`.** We experiment with different reset intervals and have found that resetting the reward model after every 20K gradient steps (1K environment steps) to work well. We hypothesize that the periodic resetting procedure might have alleviated the overfitting issue of the reward function, allowing it to continuously adapt to incoming transitions.

## C.3 Full COG results

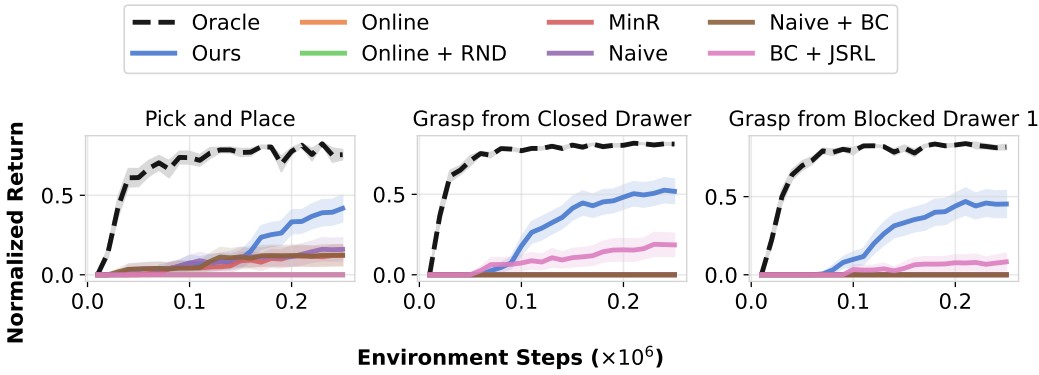

Figure 15: The performance on 3 COG tasks.

## C.4 Implicit-Q-learning (IQL) as the backbone RL algorithm.

Conceptually, our proposed method, ExPLORe, is not specific to the selected RL method, and can be combined with any RL methods that take in both offline and online transitions. Other than RLPD, we experimented with another RL algorithm, IQL [Kostrikov et al., 2021], in the offline to online fine-tuning setting where we run IQL on an online replay buffer prepended with the reward-relabelled offline data. In our experiments (Figure 16), we have found the use of the RND bonus in the offline data to be essential for accelerating learning in both medium datasets (without RND bonus, none of the runs succeed).

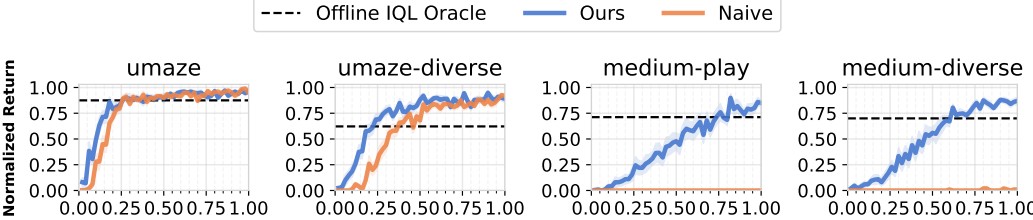

Figure 16: Results on 4 AntMaze datasets. The oracle offline IQL numbers are directly taken from the original paper. It uses the offline dataset with ground truth reward. The other curves do not have access to the ground truth reward in the offline dataset. 10 seeds are used for each curve.

## C.5 Pre-trained goal-conditioned (GC)-IQL skills.

To get a sense of how well skill-discovery may perform, we perform an additional empirical study in `antmaze-large-diverse-v2` with a set of unsupervised, pre-trained skills. In particular, we specify the skills to be a set of pre-trained goal-conditioned (GC) policies from the offline data where the 2-D goal coordinate specifies the skill. We chose this more manually defined skill set with more inductive biases to demonstrate an approximate upper-bound on an approach that discovers skills in an unsupervised fashion.

We use the offline data to pre-train goal-conditioned policies using GC-IQL on the offline dataset. In the online phase, we perform skill-inference on this set of skills by using RLPD to train a high-level policy that outputs actions corresponding to 2D goal coordinate in the maze for the underlying skill to reach. While this combination leads to increased coverage during early exploration, we found that skill inference using RLPD eventually gets stuck if reward is not found and ceases to explore further, unless additional online RND bonuses are applied. In Figure 17, we plot the performance of this skill inference comparison (5 seeds for skill-inference, 3 for skill pre-training, and ten seeds for the rest) – while inference in the space of skills leads to much faster exploration than with standard

online RND, we found the best variant of skill-discovery to be slightly worse than our method (GC-IQL+Online RLPD + Online RND vs. Ours).

In summary, we found that while skill-learning and online skill inference does lead to faster exploration over online-only exploration, it still lags in performance compared to our approach. We note that our method has several advantages over a skill-learning baseline, such as not needing a separate pre-training phase, and not needing to specify or discover a separate skill set.

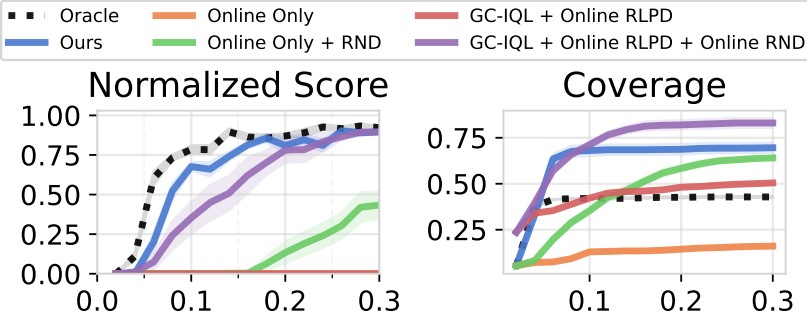

Figure 17: Results on AntMaze-Large-Diverse using pre-trained goal-conditioned IQL on offline data. The $x$-axis is the number of environment steps $\times 10^6$. The $y$-axis is the normalized return (left) and the coverage percentage (right). The pre-trained IQL agent achieves around 50% success rate by using the correct goal.

# D   Visualization Details for Figure 5

Since the prior data on COG contains no full trajectories that complete the task, to generate Figure 5, a successful trajectory on the `Grasp from Closed Drawer` environment was made by combining a successful trajectory opening the drawer and a successful trajectory taking the object out of the drawer from the prior dataset. Afterward, we evaluate the reward model on this successful trajectory for 20 training seeds and normalize each to have zero mean and unit variance.

Below, we provide the same visualization as in Figure 5, but on other successful trajectories combined from the prior data on `Grasp from Closed Drawer` and observe a similar effect.

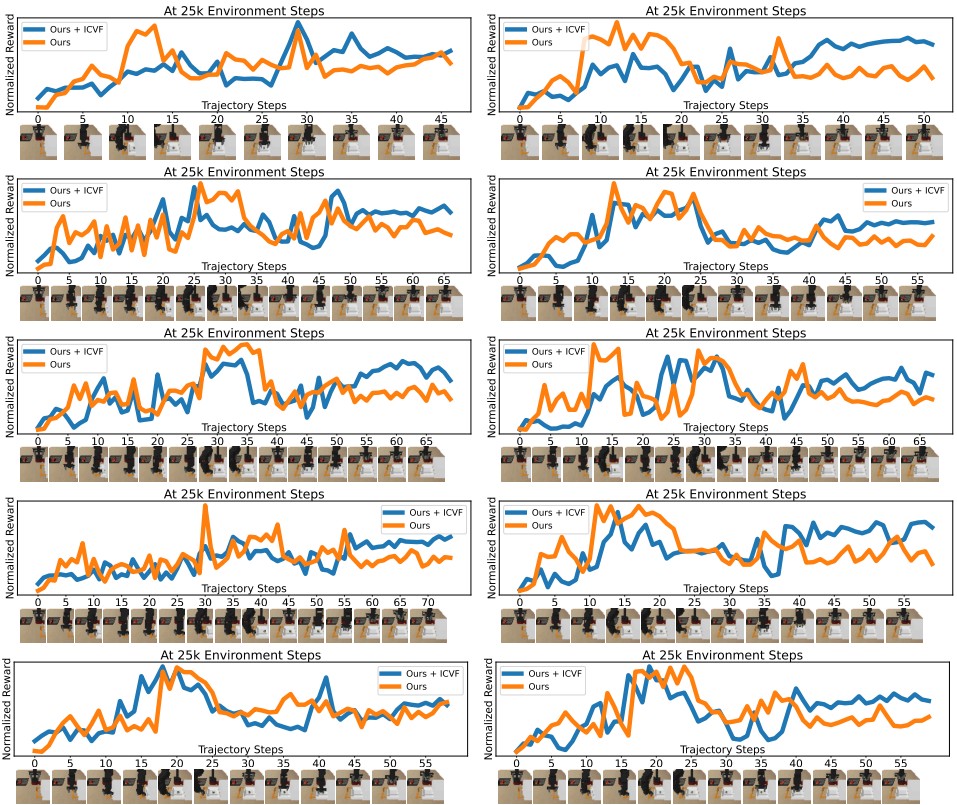

# E  Compute Description

We use Tesla V100 GPU for running the experiments. Each AntMaze experiment (300K environment steps) takes around 2 GPU hours. Each Adroit experiment (1M environment steps) takes around 6 GPU hours. Each COG experiment (250K environment steps) takes around 3 GPU hours. To reproduce all the results in the main body of the paper, it takes around 10 (seeds) $\times$ 6 (tasks) $\times$ 6 (methods) $\times$ 2 (AntMaze) + 10 (seeds) $\times$ 3 (tasks) $\times$ 6 (methods) $\times$ 6 (Adroit) + 20 (seeds) $\times$ 3 (tasks) $\times$ 6 (method) $\times$ 3 (COG) = 2880 GPU hours.

