# OpenReview forum: "Accelerating Exploration with Unlabeled Prior Data"
_NeurIPS.cc/2023/Conference — NeurIPS 2023 poster_

### Official Review · Reviewer_vsqs · 2023-06-19

**Soundness:** 4 excellent
**Presentation:** 4 excellent
**Contribution:** 4 excellent
**Rating:** 6
**Confidence:** 4

**Summary:**

This work presents an off policy RL algorithm that use prior unlabeled data in a novel manner. The mechanism use an "optimistic" reward labeled based on  a UCB estimate, that it is practically an iterative RND full update of the unlabeled dataset based on a stepwise exploration update of the RND functions. This reward of the unlabeled data smoothly decay making the agent explore better in comparison to other methods.

**Strengths:**

The paper is clear on explaining all the methods, algorithms and theory used. Has a in depth citation of the literature. Compare the methods to relevant methods. It also characterize well were the method stand with respect to related efforts. The results clearly surpass comparative methods in sample efficiency. It also re-use state of the art results (RLPD and ICVF ) in the project. It also clearly and unambiguously answer it three research questions.
Novel method for reward labeling prior unlabeled data.

**Weaknesses:**

The re-initialization of the reward model after 20k for the relocate task is puzzling, which imply that this algorithm need a more robust extension to deal with this situation. The situation is not clearly understood. Why the reset are really needed? How can the process automatic identify the needs for reset or change the interpolation dynamics of the algorithm too adapt to this situation. Seems that something is missing.
It seems that this approach it is inefficient  when processing or bad demonstration or subpar-demosntration in the sense of what we can derive from not able to learn from high dimensional data, and the need for ICVF to learn a compact state. Which is reasonable but also imply that it would been nice to have this analyzed in the work. The analysis of the quality and quantity of the unlabeled data, for example, as was done in (How to Leverage Unlabeled Data in Offline Reinforcement Learning. Yu et al 22) This might indicate that the method needs a certain type of unlabeled data to be successful.

**Questions:**

Given the concern of the previous section, the research questions might have been extended to understand the quality of the unlabeled data.  Can the authors help me understand if this is valid concern?

Other comments were done in the previous section.


**Limitations:**

None.

---

> ### Author Rebuttal · Authors · 2023-08-10
>
> Thank you for your kind words and thoughtful feedback about our paper. We addressed your main concern regarding the reward resetting by providing more insights on the nature of the underlying issue (that is likely orthogonal to our main focus of the paper) as well as providing additional empirical evidence that our method is robust against offline datasets with different qualities.
>
> ## “Why is reset really needed?”
> We hypothesize that periodical reset reduces the plasticity of the reward function and allows the reward function to be fitted to new transitions well, rather than being over-fitted to the earlier transitions observed. This plasticity issue can be especially problematic in the relocate task because the success rate is almost zero throughout the first 300K environment steps and the reward function can overfit to a constant reward target 0 (since the task is of sparse reward). The plasticity phenomenon has been observed in various prior works [1, 2, 3] in the context of deep reinforcement learning especially in the high UTD (update-to-data) regime which we study. Addressing this issue is a very important research question in general for deep RL algorithms, but is orthogonal to our goal in the paper which primarily focuses on how the reward should be labeled in offline data so that deep RL algorithms can make use of the unlabeled offline data effectively.
>
> ## "Data quality analysis”
> In addition, we provide empirical evidence that our method can be robust to different qualities of the data, which is an indirect piece of evidence that the issue underlying reward resetting might not be coming from the data qualities (Figure 1 in the one-page PDF). Specifically, we experimented with three different settings:
> - **Orthogonal Transitions**: all the transitions that move in the up/right (towards the goal) are removed from the offline data.
> - **Insufficient Coverage**: all the transitions around the goals are removed from the offline data.
> - **1% Data**: A random 1% of the transitions are kept in the offline data.
>
> Naively using our method in the **Insufficient Coverage** setting yields zero success, but this is to be expected as our algorithm does not incentivize exploration beyond the offline data. We find that this can be patched by simply combining our method with an additional RND bonus on online data, allowing our method to learn even when data has poor coverage and no data near the goal is unavailable. Similarly, we have found that our method can also boost the performance in the **Orthogonal Transitions**, where no transitions move in the direction of the goal. For the **1% Data** setting, our method without the online RND bonus can still largely match the oracle performance. It is also worth noting that in the **Orthogonal Transitions** setting, our method achieves good performance whereas the oracle fails to succeed. These additional results, while initially surprising even to us (especially the “orthogonal” setting, which seems very difficult), strongly suggest that our method can lead to significant improvement even when the data is not very good – certainly such data would be woefully inadequate for regular offline RL or naive policy initialization.
>
> In summary, although our method does rely on the coverage assumption of the offline data to achieve good performance (as pointed by reviewers and evidenced in our additional experiments), it can be easily combined with an online exploration bonus to address the coverage issue and make use of different offline data distributions effectively and robustly to accelerate online exploration.
>
> [1] Nikishin, Evgenii, et al. "The primacy bias in deep reinforcement learning." International conference on machine learning. PMLR, 2022.
>
> [2] Lyle, Clare, et al. "Understanding plasticity in neural networks." arXiv preprint arXiv:2303.01486 (2023).
>
> [3] Sokar, Ghada, et al. "The dormant neuron phenomenon in deep reinforcement learning." arXiv preprint arXiv:2302.12902 (2023).

---

### Official Review · Reviewer_hFD7 · 2023-07-03

**Soundness:** 3 good
**Presentation:** 4 excellent
**Contribution:** 3 good
**Rating:** 6
**Confidence:** 4

**Summary:**

The paper proposes a simple method to leverage reward-free data in order to accelerate exploration in sparse reward tasks. This is done by learning a reward model and an uncertainty model during online learning, labeling the unlabeled data with an upper confidence bound based on these models, and sampling from both the online data and the pseudo-labeled offline data during learning. It results in fast achievement of sparse-reward tasks in a number of domains.

**Strengths:**

- The presentation is very well done.
- The method is extremely simple and clear to follow, simple enough to implement just from the paper with some experience in the field.
- The coverage results in Figure 2 are very strong (if not so surprising).
- The simple descriptions of baselines in 4.2 are helpful.
- The results against baselines are very strong, even when compared to a topline oracle (Figure 3, 4)

**Weaknesses:**

- One of the core assumptions of the method is somewhat obscured, namely that the offline data must contain good coverage in order for learning to be accelerated, otherwise there would be a significant gap between the frontier of the offline data, and the reward that must eventually be achieved, which would still result in slow exploration. As a result, I'm not sure I agree with the phrasing in section 4.5 that this method has better exploration, thus it achieves higher reward. Good exploration in the offline data is a prerequisite rather in order for the method to work well. If that exploration were not achievable in the first place, we could not use the existing method to accelerate learning. In some sense, we must have a working policy in order to learn a working policy.
- Related work comes at the end, in general this makes it difficult to contextualize the work done in the paper against prior methods, though details on the baselines help somewhat.

**Questions:**

With respect to the first weakness above, it might be nice to have an ablation experiment that limits the frontier of the offline data so as to examine the performance in that setting . In particular this is easy with AntMaze, one can only keep transitions in the first few quadrants of the maze.

**Limitations:**

The discussion section (6) mentions limitations in the form of quirks in reward optimization on Adroit domain, as well as a hypothetical limitation due to stability in learning (which does not seem to manifest). But, as discussed in weaknesses, I think there is a large limitation in that this method as demonstrated requires coverage in the demonstrations in order to be feasible.

---

> ### Author Rebuttal · Authors · 2023-08-10
>
> Thank you for your kind words and thoughtful feedback about our paper. We addressed your main concern regarding the coverage assumption of the offline data by conducting additional ablation experiments with different offline data distributions (e.g., poor data coverage, orthogonal trajectories) and showed that our method is also amenable and robustly well-performing in these situations (Figure 1 in the one-page PDF attached).
>
> ## “It might be nice to have an ablation experiment that limits the frontier of the offline data”
> From the additional ablation experiments, we have found that our method (when combined with online exploration bonus) can effectively utilize offline data with varying data qualities (Figure 1 in the one-page PDF). Specifically, we experimented with three different settings:
> - **Orthogonal Transitions**: all the transitions that move in the up/right (towards the goal) are removed from the offline data.
> - **Insufficient Coverage**: all the transitions around the goals are removed from the offline data.
> - **1% Data**: A random 1% of the transitions are kept in the offline data.
>
> Naively using our method in the **Insufficient Coverage** setting yields zero success, but this is to be expected as our algorithm does not incentivize exploration beyond the offline data. We find that this can be patched by simply combining our method with an additional RND bonus on online data, allowing our method to learn even when data has poor coverage and no data near the goal is unavailable. Similarly, we have found that our method can also boost the performance in the **Orthogonal Transitions**, where no transitions move in the direction of the goal. For the **1% Data** setting, our method without the online RND bonus can still largely match the oracle performance. It is also worth noting that in the **Orthogonal Transitions** setting, our method achieves good performance whereas the oracle fails to succeed. These additional results, while initially surprising even to us (especially the “orthogonal” setting, which seems very difficult), strongly suggest that our method can lead to significant improvement even when the data is not very good – certainly such data would be woefully inadequate for regular offline RL or naive policy initialization.
>
> In summary, although our method does rely on the coverage assumption of the offline data to achieve good performance (as pointed by reviewers and evidenced in our additional experiments), it can be easily combined with an online exploration bonus to address the coverage issue and make use of different offline data distributions effectively and robustly to accelerate online exploration.

---

> > ### Comment · Reviewer_hFD7 · 2023-08-14
> >
> > I thank the authors for the additional experiment. It appears to me that **Insufficient Coverage** matches my intuition. I am not very surprised by the result in **Orthogonal Transitions**. This is because the demonstrations in Antmaze have random starting and ending points throughout the maze, so it is possible that even if transitions do not move toward the downstream goal, they will be moving near the downstream goal in the offline data. As the reward isn't labeled in these transitions, the direction may not be as important as the coverage. It's good to see that the addition of RND can patch some of the issues associated with a poor frontier, though the same could be argued of many other methods.
> >
> > Though I appreciate the authors' new experiment and I think it completes the presentation, I believe the core issue of the method still remains.

---

### Official Review · Reviewer_egZc · 2023-07-05

**Soundness:** 4 excellent
**Presentation:** 3 good
**Contribution:** 2 fair
**Rating:** 5
**Confidence:** 3

**Summary:**

The paper propose a method exploiting unlabeled prior data (i.e. trajectories without the reward signal) for more efficient online learning. To this end, a reward model is trained (based on the online data) and the so-labeled offline data are utilised jointly with the offline data.


**Strengths:**

The paper tackles an important problem of using prior data towards accelerating online learning. The experimental setup covers three different domains (AntMaze, Adroit, COG) and $7$ baseline methods, both of which make a solid impression. The presentation of the method is very clean, and the paper is easy to follow.

**Weaknesses:**

I find the paper's analysis somewhat shallow with respect to the design choices and analysis. Below, I list a tentative questions and suggestions:
- the paper does not study much how does the offline data distribution affects the training. I acknowledge that different domains have different characteristics with this respect, however, it is hard to conclude something meaningful here.
-  The conventional knowledge that the offline data should be treated with pessimism, here we assume optimism? How does these two relate?
- Is the solution specific to the selected offline RL method? I understand that this can be done easily, but would it work? I'd be great to have some experiments or at least some arguments.
- I'd be great to know more about the uncertainty modelling. How accurate is it? What is the best balance of the uncertainty bonus and the learned reward signal.



I am aware of practical constraints, and I do not expect the authors to include all suggestion listed above. Nevertheless, I hope that including some of them would enrich the paper.

### Small
- The number of seeds (4-5) is rather small for the RL experiments, but is perhaps excusable by number of environments and baselines.
- I am not fully convinced by Sec 4.5. It states that the method improves the coverage (a proxy of the exploration performance). However, it is not completely clear, that this yields benefits to the online training. That is, I could imagine that there is some spurious correlation here. I admit, that I do not have any particular better idea, though.
- some hacks are needed (e.g. resetting reward model)
- I find Sec 4.4 (representation learning) a little bit detached from the main topic of the paper.

**Questions:**

(see above)

**Limitations:**

Limitations are mostly missing. In the discussion section, the authors admit that the presented approach is the 'initial instantiation' and least some problems. However, it is not clear if the presented list is comprehensive.

---

> ### Author Rebuttal · Authors · 2023-08-10
>
> Thank you for your kind words and thoughtful feedback about our paper. We addressed one of your main concerns on “how the effect of offline data distribution affects training” by conducting additional ablation experiments with different offline data distributions (e.g., poor data coverage, orthogonal trajectories) and showed that our method is also amenable and robustly well-performing in these situations (Figure 1). We have also addressed another main concern on the applicability of our approach to other RL methods by showing that our method can also be applied to IQL empirically (Figure 2) and conceptually to any RL methods that can make use of both offline and online transitions.
> ## “The paper does not study much how does the offline data distribution affect the training.”
> From the additional ablation experiments, we have found that our method (when combined with online exploration bonus) can effectively utilize offline data with varying data qualities (Figure 1 in the one-page PDF). Specifically, we experimented with three different settings:
> - **Orthogonal Transitions**: all the transitions that move in the up/right (towards the goal) are removed from the offline data.
> - **Insufficient Coverage**: all the transitions around the goals are removed from the offline data.
> - **1% Data**: A random 1% of the transitions are kept in the offline data.
>
> Naively using our method in the **Insufficient Coverage** setting yields zero success, but this is to be expected as our algorithm does not incentivize exploration beyond the offline data. We find that this can be patched by simply combining our method with an additional RND bonus on online data, allowing our method to learn even when data has poor coverage and no data near the goal is unavailable. Similarly, we have found that our method can also boost the performance in the **Orthogonal Transitions**, where no transitions move in the direction of the goal. For the **1% Data** setting, our method without the online RND bonus can still largely match the oracle performance. It is also worth noting that in the **Orthogonal Transitions** setting, our method achieves good performance whereas the oracle fails to succeed. These additional results, while initially surprising even to us (especially the “orthogonal” setting, which seems very difficult), strongly suggest that our method can lead to significant improvement even when the data is not very good – certainly such data would be woefully inadequate for regular offline RL or naive policy initialization.
>
> In summary, although our method does rely on the coverage assumption of the offline data to achieve good performance (as pointed by reviewers and evidenced in our additional experiments), it can be easily combined with an online exploration bonus to address the coverage issue and make use of different offline data distributions effectively and robustly to accelerate online exploration.
> ## “The conventional knowledge that the offline data should be treated with pessimism, here we assume optimism? How does these two relate?”
> These ideas are mutually synergistic: the conventional wisdom says that points outside the offline data should be treated with pessimism (e.g. be pessimistic about any action or state not in the offline dataset). Our principle is that points within the offline data should be treated with optimism (e.g. be optimistic about any state that was in my offline dataset, but whose reward I don’t know yet). Pessimism outside the offline data encourages the model to stay within the support of the offline data, while optimism within the offline data encourages the model to explore all possible directions present in the offline data.
> ## "Is the solution specific to the selected offline RL method?”
> The solution is not specific to the selected RL method. Conceptually, our method can be combined with any RL methods that take in both offline and online transitions. For the rebuttal, we experimented with the IQL fine-tuning setting where we run IQL on an online replay buffer prepended with the reward-relabelled offline data. In our experiments (see Figure 2 in the one-page PDF attached), we have found the use of the RND bonus in the offline data to be essential for accelerating learning in both medium datasets (without RND bonus, none of the runs succeed). We will add the additional experiments in our next revision.
> ## “How accurate [is the uncertainty modeling]?
> What is the best balance of the uncertainty bonus and the learned reward signal?”
> We measure the quality of the RND estimates by measuring the coverage and performance gains from using the reward bonus; this matches how prior works on exploration bonuses (e.g., the original RND paper) have evaluated the quality of learned uncertainty estimates. If you had in mind a particular metric for measuring the quality of learned RND uncertainty estimates, we would be happy to analyze that metric in our next revision. Regarding the weighting between reward and uncertainty bonus, we found the algorithm was relatively stable to this hyperparameter in our tasks (we use a coefficient of 1), but more careful tuning may improve further.
>
> ## Small Points
> - “The number of seeds is small” – we will run more seeds (10) and include the results in our next revision.
> - Coverage as a performance proxy is not convincing – it is true that coverage does not always correlate with the performance, but having a decent amount of coverage is a prerequisite for the method to achieve good performance when the reward function is unknown (the agent needs to first discover the goal then practices reaching the goal robustly). In most of our experiments on AntMaze the baselines achieve near 0 coverage, which hinder their ability to even locate the goals.
> - “some hacks are needed” – due to the character limit, please see our response to Reviewer vsqs (the section with the heading “Why is reset really needed?”) regarding the reset.

---

> > ### Comment · Reviewer_egZc · 2023-08-13
> > **Thank you**
> >
> > Thank you for your answers and additional experiments. I am remain positive about this work and keep my score.

---

### Official Review · Reviewer_Mcz4 · 2023-07-06

**Soundness:** 3 good
**Presentation:** 2 fair
**Contribution:** 3 good
**Rating:** 6
**Confidence:** 4

**Summary:**

This paper proposes a method to leverage unlabelled offline data to expedite online exploration. The idea is to train a reward model and random network distillation models on online data and use them to construct a UCB estimate for offline data. By training an off-policy RL algorithm on both online and relabelled offline data, the agent is encouraged to visit novel states represented in the offline data that are deemed to have high rewards. This is in contrast with most online exploration methods which only expand the exploration frontier. The authors validate their method on state-based and visual domains and demonstrate that their method achieves more efficient exploration compared to baselines.

**Strengths:**

- The method is simple and effective, providing an appealing means to leverage unlabeled offline datasets to facilitate online learning.
- The experiments are substantiative, demonstrating the superiority of UCB rewards compared to pure online exploration as well as other exploration methods that utilize unlabelled offline data. It is nice to know that in addition to state-based domains, the method is also compatible with visual domains through pre-trained visual representation, which further expands its applicability. Moreover, the analytical experiments in Section 4.5 offer clear evidence that the reason behind improved online learning is efficient exploration.
- The overall presentation of the paper is coherent.

**Weaknesses:**

- The problem setting is a bit awkward. Since the offline and online data have to be from the same domain, this method does not benefit from Internet-scale unlabeled data. It makes sense if the offline data is collected from a diverse set of tasks with the same dynamics but different underlying reward functions, and the goal is to show fast adaptation to new rewards via efficient exploration. But in that case, it would have to compare with the unsupervised skill-learning line of work, e.g. DIAYN [1], RaMP [2], and I'm not sure if exploration can outperform adaptation in terms of sample efficiency.
- The quality of the offline dataset is important. Not only would the data need to be from the same domain, but it would also need to cover interesting trajectories for the task to really facilitate exploration. If the data is from a completely orthogonal task then it might even hinder exploration, depending on the balance between the reward estimate and the uncertainty quantification.

[1] Benjamin Eysenbach, Abhishek Gupta, Julian Ibarz, Sergey Levine. Diversity Is All You Need: Learning Skills Without A Reward Function. ICLR 2019.
[2] Boyuan Chen, Chuning Zhu, Pulkit Agrawal, Kaiqing Zhang, Abhishek Gupta. Self-Supervised Reinforcement Learning that Transfers using Random Features, ArXiv Preprint.

**Questions:**

- How does the quality of offline data affect performance? What happens when the offline data doesn't cover high-return trajectories? What happens if it contains orthogonal trajectories (e.g. the task is to go left, but the data goes right)? It would be helpful to see an ablation experiment with different dataset qualities vs. exploration efficiency.
- It would be nice to include a comparison to an unsupervised skill-learning baseline. I expect UCB exploration to achieve better asymptotic performance than unsupervised skill-learning when the dataset does not cover optimal trajectories.


**Limitations:**

The authors adequately addressed the limitations of their method in the paper. They mention that sometimes naively fitting the reward to online exploration can lead to overfitting. Moreover, the UCB estimates can be volatile at the beginning of training.

---

> ### Author Rebuttal · Authors · 2023-08-10
>
> Thank you for your kind words and thoughtful feedback about our paper. We addressed your first concern on offline data quality by conducting additional ablation experiments with different offline data distributions (e.g., poor data coverage, orthogonal trajectories) and showed that our method is also amenable and robustly well-performing in these situations. We addressed your second concern on the lack of unsupervised skill-learning comparison by adding an additional baseline that makes use of pre-trained goal-conditioned policies as skills and runs online RL to select among these skills.
> ## “How does the quality of offline data affect performance?”
> From the additional ablation experiments, we have found that our method (when combined with online exploration bonus) can effectively utilize offline data with varying data qualities (Figure 1 in the one-page PDF). Specifically, we experimented with three different settings:
> - **Orthogonal Transitions**: all the transitions that move in the up/right (towards the goal) are removed from the offline data.
> - **Insufficient Coverage**: all the transitions around the goals are removed from the offline data.
> - **1% Data**: A random 1% of the transitions are kept in the offline data.
>
> Naively using our method in the **Insufficient Coverage** setting yields zero success, but this is to be expected as our algorithm does not incentivize exploration beyond the offline data. We find that this can be patched by simply combining our method with an additional RND bonus on online data, allowing our method to learn even when data has poor coverage and no data near the goal is unavailable. Similarly, we have found that our method can also boost the performance in the **Orthogonal Transitions**, where no transitions move in the direction of the goal. For the **1% Data** setting, our method without the online RND bonus can still largely match the oracle performance. It is also worth noting that in the **Orthogonal Transitions** setting, our method achieves good performance whereas the oracle fails to succeed. These additional results, while initially surprising even to us (especially the “orthogonal” setting, which seems very difficult), strongly suggest that our method can lead to significant improvement even when the data is not very good – certainly such data would be woefully inadequate for regular offline RL or naive policy initialization.
>
> In summary, although our method does rely on the coverage assumption of the offline data to achieve good performance (as pointed by reviewers and evidenced in our additional experiments), it can be easily combined with an online exploration bonus to address the coverage issue and make use of different offline data distributions effectively and robustly to accelerate online exploration.
> ## “It would be nice to include a comparison to an unsupervised skill-learning baseline.”
> To get a sense of how skill-discovery may perform, we perform an additional empirical study in AntMaze-large-diverse-v2 with a set of unsupervised, pre-trained skills. In particular, we specify the skills to be a set of pre-trained goal-conditioned (GC) policies from the offline data where the 2-D goal coordinate specifies the skill. We chose this more manually defined skill set with more inductive biases to demonstrate an approximate upper-bound on an approach that discovers skills in an unsupervised fashion.
>
> We use the offline data to pre-train goal-conditioned policies using GC-IQL on the offline dataset. In the online phase, we perform skill-inference on this set of skills by using RLPD to train a high-level policy that outputs actions corresponding to 2D goal coordinate in the maze for the underlying skill to reach. While this combination leads to increased coverage during early exploration, we found that skill inference using RLPD eventually gets stuck if reward is not found and ceases to explore further, unless additional online RND bonuses are applied. In Figure 3, we plot the performance of this skill inference comparison (5 seeds for skill-inference, 3 for skill pre-training, and ten seeds for the rest) – while inference in the space of skills leads to much faster exploration than with standard online RND, we found the best variant of skill-discovery to be slightly worse than our method (GC-IQL+Online RLPD + Online RND vs. Ours).
>
> In summary, we found that while skill-learning and online skill inference does lead to faster exploration over online-only exploration, it still lags in performance compared to our approach. We note that our method has several advantages over a skill-learning baseline, such as not needing a separate pre-training phase, and not needing to specify or discover a separate skill set. We note that due to the rebuttal time limit, we were only able to tune a limited set of hyperparameters such as the horizon length of the skill, so it is entirely possible that further tuning of skill-discovery might yield stronger results.
> ## “The problem setting is awkward.”
> This is definitely a valid concern and making good use of Internet-scale unlabeled data is a great direction for future work!! There is however considerable interest in developing algorithms that can effectively utilize offline data from the same domain (e.g., [1, 2]). In addition, the problem setting is more suitable for tasks where large-scale data are not available (e.g., inventory management [3], nuclear fusion [4]).
>
> [1] Ajay, Anurag, et al. "Opal: Offline primitive discovery for accelerating offline reinforcement learning." arXiv preprint arXiv:2010.13611 (2020).
>
> [2] Pertsch, Karl, Youngwoon Lee, and Joseph Lim. "Accelerating reinforcement learning with learned skill priors." Conference on robot learning. PMLR, 2021.
>
> [3] Madeka, Dhruv, et al. "Deep inventory management." arXiv preprint arXiv:2210.03137 (2022).
>
> [4] Degrave, Jonas, et al. "Magnetic control of tokamak plasmas through deep reinforcement learning." Nature 602.7897 (2022): 414-419.

---

> > ### Comment · Reviewer_Mcz4 · 2023-08-15
> >
> > Thank you for running additional experiments and addressing my questions. I appreciate the antmaze results with insufficient coverage, which match my intuition, and the comparisons to unsupervised skill learning, which is a bit surprising. There might be a misunderstanding of the term "orthogonal data," though. What I meant is that the data **coverage** is orthogonal to the task, rather than the **transitions**. For example, consider a crossroad where the goal is to go straight but the data only covers turning left and right. In this case, an exploration bonus towards the left and the right might slow down the task-relevant exploration. It is rather intuitive that the proposed method performs well with orthogonal transitions since the UCB is defined on state-action pairs and does not involve transitions. That said, I think the experiments with insufficient coverage have sufficiently made the point, and the unsupervised skill learning results reveal further potential of the method. I will therefore keep my positive score.

---

### Author Rebuttal · Authors · 2023-08-10

We appreciate the feedback from all reviewers. We added three additional sets of experimental results in the one-page PDF attached to address the three shared main concerns from the reviewers --

**1. How does offline data quality affect online performance? -- Figure 1**

We ran additional ablation experiments with varying offline data distributions, and we have found that our method (when combined with online exploration bonus) can effectively utilize offline data with varying data qualities. Although our method alone does rely on the coverage assumption of the offline data to achieve good performance (as pointed by reviewers and evidenced in our additional experiments), it can be easily combined with an online exploration bonus to address the coverage issue and make use of different offline data distributions effectively and robustly to accelerate online exploration.

**2. Can our method be used in other RL algorithms? -- Figure 2**

Conceptually, our method can be combined with any RL methods that take in both offline and online transitions. To illustrate this point, we ran additional experiments where we applied our method on top of IQL where we run IQL on an online replay buffer prepended with the reward-relabelled offline data. In our experiments, we have found the use of the RND bonus in the offline data to be essential for accelerating learning in both medium datasets (without RND bonus, none of the runs succeed).

**3. Lack of unsupervised skill discovery baseline. -- Figure 3**

We include an additional baseline on the AntMaze-large-diverse-v2 with a set of unsupervised, pre-trained skills. In particular, we specify the skills to be a set of pre-trained goal-conditioned (GC) policies from the offline data where the 2-D goal coordinate specifies the skill. We chose this more manually defined skill set with more inductive biases to demonstrate an approximate upper-bound on an approach that discovers skills in an unsupervised fashion. We found that while skill-learning and online skill inference does lead to faster exploration over online-only exploration, it still lags in performance compared to our approach.

Please see our individual responses for more details on these experiments and answers to individual questions.

---

### Decision · Program_Chairs · 2023-09-21

**Decision:**

Accept (poster)

**Comment:**

This submission is about a method leveraging unlabelled offline data for efficient online model-free RL exploration. The proposed method is very simple and intuitive, yet showed good deal of improvement over baseline methods. Of course, the problem the submission deals with is a timely important problem too. So, the reviewers all suggested to accept the submission. However, there were a few concerns regarding limited evaluation domains and also its applicable problem setup. So, we recommend authors to further add more experiments during the camera ready preparation period if possible.